# Mind the Graph When Balancing Data for Fairness or Robustness

**Jessica Schrouff**
Google DeepMind
schrouff@google.com

**Alexis Bellot**
Google DeepMind

**Amal Rannen-Triki**
Google DeepMind

**Alan Malek**
Google DeepMind

**Isabela Albuquerque**
Google DeepMind

**Arthur Gretton**
Google DeepMind
Gatsby, UCL

**Alexander D'Amour**
Google DeepMind

**Silvia Chiappa**
Google DeepMind

## Abstract

Failures of fairness or robustness in machine learning predictive settings can be due to undesired dependencies between covariates, outcomes and auxiliary factors of variation. A common strategy to mitigate these failures is data balancing, which attempts to remove those undesired dependencies. In this work, we define conditions on the training distribution for data balancing to lead to fair or robust models. Our results display that, in many cases, the balanced distribution does not correspond to selectively removing the undesired dependencies in a causal graph of the task, leading to multiple failure modes and even interference with other mitigation techniques such as regularization. Overall, our results highlight the importance of taking the causal graph into account before performing data balancing.

## 1 Introduction

When training prediction models, practitioners often desire that the model's outputs display safety properties in addition to high performance, such as being fair across demographic subgroups [29, 51] or being robust to distribution shifts [e.g. 19, 59]. These objectives can be difficult to attain if there are undesired dependencies between covariates $X$, labels $Y$, and auxiliary factors of variation $Z$, such as confounding factors or hidden stratification [26, 27]. A commonly referenced example is that of an animal classification task from wildlife pictures [e.g. 64]: the model might identify patterns in the background of the images that are indicative of the type of animal (e.g. the presence of snow for polar bears or grass for cows), which might lead to the model failing to recognize the same animal when it is on another background. When the auxiliary factors relate to demographic attributes, the deployment of such models can have societal implications, e.g. patients not being assigned medical resources due to factors related to race [54].

Multiple mitigation strategies have been proposed to remove undesired dependencies pre-, in- or post-processing. Amongst them, balancing the training data is typically considered a straightforward approach and has been used or researched in various settings [e.g. 37, 39, 60, 8, 33, 40, 2]. This approach modifies the training distribution, indicated with $P^t(X, Y, Z)$, into a new, balanced distribution (which we refer to as $Q(X, Y, Z)$) that aims to approximate an 'idealized' training distribution in which the undesired dependencies are absent [48, 14, 77]. Models are then trained on this balanced distribution to attain different fairness or robustness criteria. A popular approach to construct a balanced distribution is by balancing classes (resp. groups), leading to a uniform distribution over $Y$ (resp. $Z$). While successful for addressing failures of robustness [e.g. 33] or of fairness due to under-representation of certain groups [e.g. 75], this approach does not induce independence between

$Y$ and $Z$. To approximate independence, a 'joint' balancing on $(Y, Z)$ is often performed [e.g. 48, 8]. Joint balancing can be implemented by matching the numbers of samples in all $(y, z)$ groups (only feasible when $Y$ and $Z$ have small, discrete domains) via subsampling the majority groups [e.g. 8], upsampling the minority groups [e.g. 63], resampling the data with weights proportional to $P^t(Y)P^t(Z)/P^t(Y,Z)$, or reweighting the loss [9]. Our work focuses on joint balancing given its suitability to mitigate a marginal dependence between $Y$ and $Z$.[1] While the choice of the method for jointly balancing can impact the results [11, 65, 33], these methods can be all seen as modifying $P^t$ as described in Definition 1.1.

**Definition 1.1** (Jointly balanced distribution). We say that the distribution $Q(X, Y, Z)$ is a jointly balanced version of $P^t(X, Y, Z)$ if $Q(X, Y, Z) = P^t(X, Y, Z)\frac{P^t(Y)P^t(Z)}{P^t(Y,Z)}$.

In some cases, data balancing has proven to be an effective mitigation strategy for undesired dependencies, performing on-par with other, more complex mitigation techniques [33]. Recently, data balancing has also shown promises for mitigation during fine-tuning or partial retraining [41, 44, 49, 79, 75], which is relevant to the settings of training large-scale models and with large amounts of data. Nevertheless, data balancing has also displayed failure modes in which the obtained models were not fair, robust or optimal [76, 48, 58, 2]. These failure modes have not been thoroughly characterized and can be difficult to predict. Furthermore, the impact of data balancing on other mitigation strategies has not been studied extensively.

Given data balancing's popularity as a baseline mitigation strategy for undesired dependencies, we aim to formalize some of its promises and pitfalls. Our analysis relies on a causal graphical framework, which allows investigating the impact of data balancing in different data generating processes. Our contributions can be summarized as follows: (1) we display failure modes of data balancing in semi-synthetic tasks and highlight how predicting these failures can be challenging; (2) we introduce conditions for data balancing to attain invariance to undesired dependencies as defined by fairness or robustness criteria; (3) we prove that data balancing does not correspond to 'removing' undesired dependencies from a causal perspective, and can negatively impact fairness or robustness criteria when combined with regularization strategies; and (4) we illustrate how our findings can be used to distinguish between failure modes and identify next steps.

## 2 Preliminaries

Let $X$, $Y$, $Z$ be discrete random variables with $X \in \mathcal{X}$ corresponding to a set of covariates (e.g. tabular, images or text), $Y \in \mathcal{Y}$ to an outcome to be predicted, and $Z \in \mathcal{Z}$ to an auxiliary factor of variation, such as a sensitive attribute or the type of background of an image, that displays statistical dependence with $Y$. We assume access to data sampled from distribution $P^t(X, Y, Z)$, where $P^t$ is the true data-generating distribution. We consider a family of models $\mathcal{F} \in \mathcal{X} \to \mathcal{Y}$ that will be trained on data from $P^t(X, Y, Z)$ to minimize the risk $R_{P^t}(f) := \mathbb{E}_{X,Y \sim P^t}[\ell(f; X, Y)]$ where $\ell$ is a loss function. We define $f^* \in \mathcal{F}$ to be the *optimal* model, i.e. one where the risk attains the minimum on $P^t$. We assume that $\mathbb{E}_Q[Y|X] = f^*(X)$, which occurs, for example, if $\ell$ is the square loss or cross-entropy loss.

**Definition 2.1** (Optimality). A prediction model $f \in \mathcal{F}$ is optimal w.r.t. $P^t$ if $f = \text{argmin}_{f' \in \mathcal{F}} R_{P^t}(f')$.

### 2.1 Desired criteria on a model's predictions

Due to undesired independencies, while a model may be optimal on $P^t$, it might not be optimal on another distribution of interest $P'(X, Y, Z)$ (e.g. in deployment), and/or might display disparities across subsets of the data (e.g. $P^t(X, Y \mid Z = z)$) [22]. To mitigate this issue, multiple safety criteria have been defined in the fields of *fairness* and *robustness*.

**Fairness:** Fairness criteria can be defined in terms of the dependence between the model's output $f(X)$ and the auxiliary factor of variation $Z$. We consider established fairness criteria [5, 51], including *demographic parity* [$f(X) \perp\!\!\!\perp Z$, 23], *equalized odds* [$f(X) \perp\!\!\!\perp Z \mid Y$, 29] and *predictive parity* [$Y \perp\!\!\!\perp Z \mid f(X)$, 24]. Beyond fairness of $f(X)$, we also consider fairness of intermediate *representations* $\phi(X)$, e.g. $\phi(X) \perp\!\!\!\perp Z$ [81], for their usage in downstream tasks.

---

[1]We briefly discuss group or class data balancing in Appendix A.1.

**Robustness:** In this field, the focus is typically on finding models $f_\theta$ parameterized by $\theta \in \Theta$ that provide the lowest risk across a *family of target distributions* $\mathcal{P}$. For instance, the 'worst group performance' criterion aims to select parameters such that the performance on a 'worst' distribution $P'$ is optimized, i.e. $\theta^* = \min_{\theta \in \Theta}\{\sup_{P' \in \mathcal{P}} R_{P'}(f_\theta)\}$ [6, 20]. $\mathcal{P}$ can be defined so that each distribution $P'$ represents a specific subpopulation [64], to minimize the loss in each subgroup, or aiming for an invariance of $R_{P'}$ across subgroups [*risk-invariance*, 48].

**Definition 2.2** (Risk-invariance). A prediction model $f$ is risk-invariant w.r.t. a family of distributions $\mathcal{P}$ if $R_P(f) = R_{P'}(f) \, \forall P, P' \in \mathcal{P}$.

If a model is optimal on $P^t$ and risk-invariant w.r.t. $\mathcal{P}$, it is also optimal w.r.t. $\mathcal{P}$. The choice of $\mathcal{P}$ is context-specific and reflects some domain knowledge about shifts that are likely to arise in a given application. For instance, a plausible family of target distributions could imply a shift in the dependence between $Y$ and $Z$, also known as a *correlation shift* [62], and be expressed as $\mathcal{P} = \{P'(X, Y, Z) = P^t(X \,|\, Y, Z)P'(Z \,|\, Y)P^t(Y), \forall P'(Z \,|\, Y)\}$. Alternatively, we can define $\mathcal{P}$ using a causal framework (see Section 2.2) when the data generation process is known [48].

We acknowledge that selecting amongst those criteria is context-dependent and do not advocate for a specific choice. We call a prediction model $f$ *invariant* to undesired dependencies, denoted with $f \in \mathcal{F}_{inv}$, if it satisfies one of such criteria. For brevity, we focus on risk-invariance in the main text and consider fairness criteria in Appendix. Obtaining an invariant model can be performed in different ways, with data balancing being a popular approach.

## 2.2 Causal framework to analyse data balancing

To understand the effects of data balancing, we need to investigate its impact on the distribution $P^t$. A causal formalization is useful for studying how distributions change under different interventions. To analyse the implications of data balancing, we use the framework of *causal Bayesian networks* (CBNs) [e.g. 71, 13, 52, 74, 25, 48]. A Bayesian network [55, 56, 15, 42] is a pair $\langle \mathcal{G}, P^t \rangle$, in which $\mathcal{G}$ is a directed acyclic graph whose nodes $X^1, \ldots, X^D$ represent random variables and in which $P^t$ is a joint distribution over the nodes. The absence of edges in $\mathcal{G}$ implies a set of statistical independence assumptions satisfied by $P^t$ that can be expressed by the factorization $P^t(X^1, \ldots, X^D) = \prod_{d=1}^{D} P^t(X^d \,|\, \mathrm{pa}(X^d))$, where $\mathrm{pa}(X^d)$ denote the *parents* of $X^d$, namely the nodes with an edge into $X^d$ (we say that $P^t$ *factorizes according to* $\mathcal{G}$). A CBN is a Bayesian network in which an edge expresses causal influence, so that $\mathrm{pa}(X^d)$ are *direct causes* of $X^d$. A directed path between $X^i$ and $X^j$ in a CBN is also called a *causal path*. A non-directed path, also called *non-causal path*, expresses statistical dependence of non-causal nature. We refer to the statistical dependence between $X^i$ and $X^j$ that arises only due to the presence of non-causal paths as *purely spurious*. In our setting $X^1 \cup \cdots \cup X^D = X \cup Y \cup Z \cup \mathbf{U}$ where $\mathbf{U}$ are unobserved variables. Inspired by prior work [74, 3, 70, 77], we make a decomposition assumption on the form of the covariates $X$.

**Assumption 2.3** (Decomposition of $X$). The covariates $X$ can be decomposed into three unobserved random variables $X_{\bar{Z}}^{\perp}, X_{\bar{Y}}^{\perp}$ and $X_{Y \wedge Z}$ such that: 1) $X_{\bar{Z}}^{\perp}$ does not have causal paths to/from $Z$ but has causal paths to/from $Y$, 2) $X_{\bar{Y}}^{\perp}$ does not have causal paths to/from $Y$ but has causal paths to/from $Z$, 3) $X_{Y \wedge Z}$ has causal paths to/from both $Y$ and $Z$, representing *entangled* signals, and 4) $X$ is measurable w.r.t. $\sigma(X_{\bar{Z}}^{\perp}, X_{\bar{Y}}^{\perp}, X_{Y \wedge Z})$, the joint $\sigma$-algebra. In particular, there exists a function $g$ such that $X = g(X_{\bar{Z}}^{\perp}, X_{\bar{Y}}^{\perp}, X_{Y \wedge Z})$ almost everywhere and $P^t(X_{\bar{Z}}^{\perp}, X_{\bar{Y}}^{\perp}, X_{Y \wedge Z}, Y, Z, \mathbf{U}) = P^t(g(X_{\bar{Z}}^{\perp}, X_{\bar{Y}}^{\perp}, X_{Y \wedge Z}), Y, Z, \mathbf{U})$.

In the animal classification example, $X_{\bar{Z}}^{\perp}$ would correspond to the animal pixels, $X_{\bar{Y}}^{\perp}$ to the background pixels (e.g. snowy or grassy landscape), and $X_{Y \wedge Z}$ to characteristics of the animal that depend on its environment (e.g. color of the fur pixels in bears). Intuitively, we want to build a prediction model $f$ that only depends on the animal pixels. While the decomposition may be readily available when a causal graph of the application is available and the data is tabular, we typically do not have direct access to the different functions of $X$ and these need to be isolated algorithmically.

Following Schölkopf et al. [66], we consider both the case in which $X_{\bar{Z}}^{\perp} \cup X_{Y \wedge Z}$ are direct causes of the label $Y$ (*causal task*) e.g. estimating the helpfulness of a text review, and the case in which $Y$ is a direct cause of $X_{\bar{Z}}^{\perp} \cup X_{Y \wedge Z}$ (*anti-causal task*) as in object detection tasks in computer vision. Figures 1(a-b) display examples of anti-causal and causal tasks with a purely spurious dependence

| | Graph | Data Balancing | Regularization | Next steps |
|---|---|---|---|---|
| (a) | $Y \rightarrow X_{\bar{Z}}^{\perp}$ $U$ $Z \rightarrow X_Y^{\perp}$ | ✓risk-invariant ✓optimal | $f(X) \perp\!\!\!\perp Z \mid Y$ ✓risk-invariant ✓optimal | N.A. |
| (b) | $Y \leftarrow X_{\bar{Z}}^{\perp}$ $U$ $Z \rightarrow X_Y^{\perp}$ | ✓risk-invariant ✗optimal | $f(X) \perp\!\!\!\perp Z$ ✓risk-invariant ✓optimal | Use regularization, without prior balancing (as per Section 5) |
| (c) | $V \rightarrow X_V$ $U_2$ $U_3$ $Y \rightarrow X_{\bar{Z}}^{\perp}$ $U_1$ $Z \rightarrow X_Y^{\perp}$ | ✗risk-invariant ✗optimal | $f(X) \perp\!\!\!\perp Z \mid Y$ ✗risk-invariant ✗optimal | Refer to Kaur et al., 2023; Alabdulmohsin et al., 2024 which address the cases with multiple auxiliary factors. |
| (d) | $Y \rightarrow X_{\bar{Z}}^{\perp}$ $U$ $Z \rightarrow X_{Y \wedge Z}$ | ✗risk-invariant ✗optimal | $f(X) \perp\!\!\!\perp Z \mid Y$ ✓risk-invariant ✓optimal | Use regularization |

Table 1: Examples of causal Bayesian networks with undesired dependencies between $Y$ and $Z$ displayed by red edges. Light gray indicates unobserved variables. $X_{Y \wedge Z} = \emptyset$ in (a-b) and there is no entanglement between $Y$ and $Z$ via $X$. In (c), we expand the system to include $V \in \mathbf{U}$ and its influence on $X$, which is given by $X_V$. For each Causal Bayesian Network considered, we display when data balancing leads to a risk-invariant and/or optimal model. We compare these with regularization following Veitch et al. [74] and suggest next steps.

between $Y$ and $Z$. It is important to note that statistical relationships between the different variables and functions of $X$ are determined by the graph: for instance, in Figure 1(a) $X_{\bar{Z}}^{\perp} \perp\!\!\!\perp Z \mid Y$, while in Figure 1(b) $X_{\bar{Z}}^{\perp} \perp\!\!\!\perp Z$.

Based on a CBN of the task and Assumption 2.3, we characterize undesired dependencies as the presence of undesired paths between $Z$ and $Y$, which we indicate through red edges (Figure 1). Based on this depiction of undesired dependencies, we can define the family of target distributions $\mathcal{P}$ such that black edges are preserved, but those in red may lead to changes in the distribution. For the anti-causal task in Figure 1(a), we can hence write $\mathcal{P} = \{P'(Y, Z, X) = P^t(Y)P'(Z \mid Y)P^t(X_{\bar{Z}}^{\perp} \mid Y)P^t(X_Y^{\perp} \mid Z)\}$ in which $P'(Z \mid Y)$ represents any distribution but all other causal mechanisms are fixed [48], which corresponds to a correlation shift.

## 3 Can we predict when data balancing fails?

As reported previously, data balancing can display failure modes, e.g. due to the presence of other confounders [76, 2], finite sampling effects [48] or a dependence between $Y$ and $Z$ when conditioning on $X$ ($Y \not\perp\!\!\!\perp Z \mid X$) [58]. However, this list is non-exhaustive and, to the best of our knowledge, there is no unifying study of those failure modes or of how they could be mitigated. In this section, we perform joint data balancing on different tasks to illustrate that successes and failures of this approach can be difficult to predict (see Table 1). For details of the experiments, see Appendix D.

Let's first consider semi-synthetic examples generated from the graphs in Figure 1(a,b), i.e. an anti-causal and causal task with a purely spurious correlation. We aim to obtain a risk-invariant and optimal model on these tasks by training on the jointly balanced distribution $Q$.

**Anti-causal task: number detection in MNIST**. Inspired by Brown et al. [8], we modify MNIST images [45, 17] by adding a factor of variation $Z$ such that the top of the image is replaced by red noise for $Z = 0$ and blue noise for $Z = 1$ (Figure 1). We sample a dataset in which the factor of variation and label are dependent ($P^t(Y = 0 \mid Z = 0) = 0.95$, $P^t(Y = 1 \mid Z = 0) = 0.10$, called the 'confounded' data), a jointly balanced dataset, and a dataset from a distribution $P^0$ in which the undesired dependency is absent ($P^0(Z = 0 \mid Y) = 0.5$). We train convolutional networks to predict whether the number in an image is smaller or larger than 5, assessing the models on their training distribution and on $P^0$.

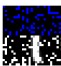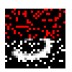
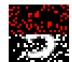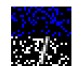

Figure 1: MNIST data samples.

Table 2: Model performance on semi-synthetic data, for the tasks in Figure 1. 'Acc' refers to accuracy, 'Worst Grp' to worst group accuracy, 'Encoding' to confounder encoding as measured by transfer learning and 'Equ. Odds' refers to equalized odds between $Z$ subgroups. $\uparrow$ (resp. $\downarrow$) means the higher (resp. lower), the better.

| Graph | Dataset | $P^t$ | $P^0$ | | | |
| | | Acc. ($\uparrow$) | Acc. ($\uparrow$) | Worst Grp ($\uparrow$) | Encoding ($\sim 0.5$) | Equ. Odds ($\downarrow$) |
|---|---|---|---|---|---|---|
| (a) | 95/10 | $0.937 \pm 0.002$ | $0.717 \pm 0.027$ | $0.380 \pm 0.062$ | $0.996 \pm 0.004$ | $0.539 \pm 0.015$ |
| (a) | Balanced | $0.871 \pm 0.008$ | $0.880 \pm 0.006$ | $0.836 \pm 0.075$ | $0.486 \pm 0.005$ | $0.018 \pm 0.008$ |
| (b) | Class bal. | $0.658 \pm 0.015$ | $0.558 \pm 0.015$ | $0.092 \pm 0.015$ | $0.690 \pm 0.113$ | $0.542 \pm 0.098$ |
| (b) | Jointly bal. | $0.574 \pm 0.016$ | $0.583 \pm 0.017$ | $0.399 \pm 0.014$ | $0.545 \pm 0.037$ | $0.060 \pm 0.046$ |
| (c) | With $V$ | $0.769 \pm 0.001$ | $0.647 \pm 0.023$ | $0.555 \pm 0.031$ | $0.665 \pm 0.134$ | $0.094 \pm 0.035$ |
| (d) | Entangled | $0.903 \pm 0.011$ | $0.672 \pm 0.004$ | $0.000 \pm 0.001$ | $0.881 \pm 0.223$ | $0.554 \pm 0.028$ |

Models trained with confounded data (95/10) display biased outputs (Table 2), with low worst group performance and high equalized odds. Performance on $P^0$ is also lower compared to that on $P^t$ ($0.937 \pm 0.002$), showing that these models are not risk-invariant w.r.t. $\mathcal{P}$. Models trained from balanced data obtain high overall performance and worst group accuracy, as well as low equalized odds. In addition, we were not able to decode $Z$ from the model representation $\phi(X)$ at the penultimate layer, suggesting that the model has not learned $X_Y^\perp$.

**Causal task: helpfulness of reviews with Amazon reviews [53]**. Inspired by Veitch et al. [74], we refer to the causal task of predicting the helpfulness rating of an Amazon review (thumbs up or down, $Y$) from its text ($X$). We add a synthetic factor of variation $Z$ such that words like 'the' or 'my' are replaced by 'thexxxx' and 'myxxxx' ($Z = 0$) or 'theyyyy' and 'myyyyy' ($Z = 1$). We train a BERT [34] model on a class-balanced version of the data for reference (due to high class imbalance), and compare to a model trained on jointly balanced data, both evaluated on their training distribution and on a distribution $P^0$ with no association.

In this case, jointly balancing improves fairness and risk-invariance, with the model's performance on the training distribution (acc.: $0.574 \pm 0.016$) being similar to that on $P^0$ (Table 2). This however comes at a high performance cost when compared to the class balanced model's performance on $P^t$ (acc: $0.658 \pm 0.015$). Therefore, data balancing might not lead to optimality for this causal task.

Using the same framework, we can replicate the failure modes due to another confounder described in Wang et al. [76], Alabdulmohsin et al. [2] as well as that from Puli et al. [58].

**Anti-causal task with another factor of variation $V$.** It is common for multiple auxiliary factors to influence the data generating process, and they tend to correlate with each other [e.g. 21]. To emulate this case, we introduce more unobserved variables $U_2, U_3$ as well as a factor of variation $V$ which affects the data through $X_V$ (Figure 1(c)). We modify the MNIST data generation to include $X_V$ depicted by a green cross on the top left or top right of the image and jointly balance the data on $(Y, Z)$ before training the model. We evaluate the obtained predictor on a distribution where $V$ and $Z$ are not correlated with $Y$ and observe (Table 2) a large gap between worst group accuracy and overall performance, as well as non-null equalized odds. These results suggest that the model is not fair or robust.

**Anti-causal task with entangled data.** We map the work in Puli et al. [58] to our decomposition of $X$ and propose the example graph in Figure 1(d) where $X_{Y \wedge Z}$ represents an entangled function of $X$. To match this data generating process, the color of the noise in MNIST samples is defined by $\text{OR}(Y, Z)$ and the evaluation distribution is the disentangled $P^0$ with no dependence between $Y$ and $Z$. Once again, the obtained model is not fair, robust or optimal (Table 2). Appendix A.2 discusses this case further.

Motivated by these examples of both success and failures, we define conditions for the success of data balancing, and highlight when the cases above fail to meet these conditions.

# 4 Conditions for data balancing to produce an invariant and optimal model

In this section, we introduce a sufficient condition on the data generative process and a necessary condition on the trained model that, taken together, lead to a risk-invariant and optimal prediction model after training on $Q$ (proofs in Appendix B.1). In Appendix B.2, we derive similar conditions

for fairness criteria. Throughout the rest of the paper, we use an subscript to indicate under which of $P^t$ or $Q$ a statistical independence holds, e.g. $Y \perp\!\!\!\perp_{P^t} Z$ to indicate $P^t(Y \mid Z) = P^t(Y)$.

We consider the criterion of risk-invariance (Definition 2.2) under correlation shift, i.e. $\mathcal{P} = \{P'(X, Y, X) = P^t(X|Y, Z)P'(Z|Y)P^t(Y)\}$. According to our decomposition of $X$, the risk-minimizing function $f(X) := \mathbb{E}_Q[Y \mid X]$ should only be a function of $X_{\bar{Z}}^{\perp}$ and not of $X_{\bar{Y}}^{\perp}$ or $X_{Y \wedge Z}$. To achieve this result with data balancing, we build on a prior result by Makar et al. [48], which shows that a model trained on a balanced distribution only depends on $X_{\bar{Z}}^{\perp}$ if $X_{\bar{Z}}^{\perp}$ represents a *sufficient statistic* for $Y$, i.e. no other part of $X$ influences $Y$.

**Definition 4.1.** (Sufficient Statistic) We say that $X_{\bar{Z}}^{\perp}$ is a sufficient statistic for $Y$ in $Q$ if $\mathbb{E}_Q[Y \mid X] = \mathbb{E}_Q[Y \mid X_{\bar{Z}}^{\perp}]$ (note that $X_{\bar{Z}}^{\perp}$ is a function of $X$).

Definition 4.1 implies that the risk-minimizing function $f$ for $Q$ does not vary with $X_{\bar{Y}}^{\perp}, X_{Y \wedge Z}$. However, this condition is not sufficient on its own to ensure that $f$ is risk-invariant w.r.t. $\mathcal{P}$, as $X_{\bar{Z}}^{\perp}$ or $Y$ may have non-causal relationships with $Z$. To ensure optimality and risk-invariance w.r.t. $\mathcal{P}$, we derive the sufficient condition in Proposition 4.2.

**Proposition 4.2.** *If $X_{\bar{Z}}^{\perp} \perp\!\!\!\perp_Q Z \mid Y$ and $X_{\bar{Z}}^{\perp}$ is a sufficient statistic for $Y$ in $Q$, then the risk-minimizer $f(X) := \mathbb{E}_Q[Y \mid X]$ is risk-invariant and optimal w.r.t. $\mathcal{P}$.*

The conditions of Proposition 4.2 concern $Q$. However, it would be of interest to express them in $P^t$ if it is possible to observe all covariates (e.g. in the case of tabular data). Based on our expression for $Q$, we can derive sufficient conditions on $P^t$, expressed in Corollary 4.3. Let's denote $\{X_{\bar{Y}}^{\perp}, X_{Y \wedge Z}\}$ by $R$.

**Corollary 4.3.** *If $R \perp\!\!\!\perp_{P^t} \{Y, X_{\bar{Z}}^{\perp}\} \mid Z$ and $X_{\bar{Z}}^{\perp} \perp\!\!\!\perp_{P^t} Z \mid Y$, then the risk-minimizer $f(X) := \mathbb{E}_Q[Y \mid X]$ is risk-invariant and optimal w.r.t. $\mathcal{P}$.*

In general, we can expect that anti-causal tasks with purely spurious correlations will satisfy these conditions, as per their definition. However, this would not be the case for most causal tasks as $X_{\bar{Z}}^{\perp} \not\perp\!\!\!\perp_{P^t} Z \mid Y$. This result is in line with our findings in Section 3, as the MNIST data generated from the graph in Figure 1(a) validates Corollary 4.3, but the Amazon reviews data generated from Figure 1(b) does not.

It may be less obvious, but the conditions for a sufficient statistic are not met in Figures 1(c,d) as $X_V \not\perp\!\!\!\perp_{P^t} \{Y, X_{\bar{Z}}^{\perp}\} \mid Z$ in the case of another factor of variation $V$, and $X_{Y \wedge Z} \not\perp\!\!\!\perp_{P^t} \{Y, X_{\bar{Z}}^{\perp}\} \mid Z$ in the case of entangled data. We hence see that when a causal graph of the application is available, Corollary 4.3 can provide indicators on when data balancing might succeed or fail, with the caveat that it is not a necessary condition.

While Proposition 4.2 and Corollary 4.3 provide conditions on the data generating process, prior work [e.g. 10, 31] has demonstrated that the learning strategy also influences the model's fairness and robustness characteristics.

**Proposition 4.4.** *Let $\hat{f} \in \mathcal{F}$ be some fitted model and $\epsilon > 0$. Assume that, for all $P', P'' \in \mathcal{P}$, we have $\left| \mathbb{E}_{P'}[Y \mid \hat{f}(X, Y)] - \mathbb{E}_{P'}[Y \mid X_{\bar{Z}}^{\perp}] \right| \leq \frac{\epsilon}{2}$. Then $\hat{f}$ is $\epsilon$-risk invariant, meaning that*

$$\sup_{P', P'' \in \mathcal{P}} R_{P'}(\hat{f}) - R_{P''}(\hat{f}) \leq \epsilon.$$

Proposition 4.4 states that the learned function $\hat{f}$ needs to be nearly optimal over $\mathcal{P}$. This statement, while straightforward, implies that (i) $\hat{f}(X)$ needs to preserve all the information about the expectation of $Y$ in $X_{\bar{Z}}^{\perp}$, and that (ii) $\hat{f}(X)$ changes with $X_{\bar{Y}}^{\perp}$ or $X_{Y \wedge Z}$ only marginally. Let's rewrite $\hat{f}(X) = h(\phi(X))$, where $h$ is 'simple' function and $\phi(X)$ is a model representation. This case could correspond to the last layers of a neural network or when learning a model based on a representation $\phi(X)$ (e.g. embeddings, transfer learning). Based on Proposition 4.4, $\phi(X)$ must be disentangled in the sense that the simple function $h$ eliminates any dependence on $X_{\bar{Y}}^{\perp}$ or $X_{Y \wedge Z}$. For example, if $h$ is a linear function, it must be possible to linearly project out all dependence on $X_{\bar{Y}}^{\perp}$ and $X_{Y \wedge Z}$. We note that such a representation can be obtained even if the data is entangled, e.g. by dropping modes of variation during training. Unlike other strategies [4, 48, 58], data balancing cannot enforce this property on its own and a disentangled representation would be necessary. This condition hence

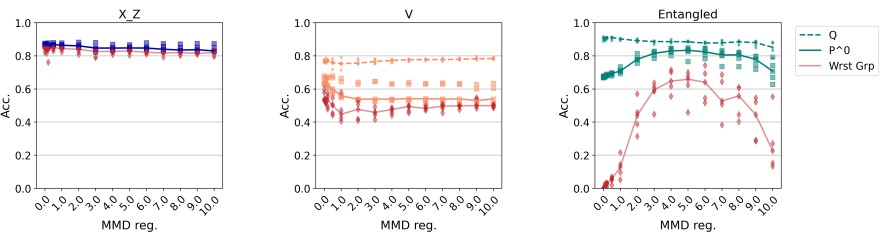

Figure 2: Accuracy across different values of the MMD hyper-parameter for models trained on balanced data and evaluated on their respective training distribution (dashed) and $P^0$ (solid line) averaged across replicates. We consider anti-causal tasks: (left) purely spurious case, (middle) when another confounder $V$ is present, and (right) the entangled dataset. Worst group performance on $P^0$ is displayed in red. Markers display individual replicates.

suggests another failure mode of data balancing when the conditions on the data are validated, but the representation is of low quality. We believe this failure mode is displayed in Kirichenko et al. [41], as the success of their data balancing mitigation only holds when using models pre-trained on large datasets.

In this section, we have identified conditions for data balancing to be successful. In the next section, we go one step further to understand how data balancing impacts the data generating process, and how it interacts with other mitigation strategies for undesired dependencies, focusing on regularization.

## 5 Impact of data balancing on the CBN

Joint data balancing is assumed to *remove* statistical dependence between $Y$ and $Z$ while keeping other relationships in the CBN of the task unaffected [e.g. 48, 77, 14]. This could be interpreted as 'dropping' edges in the undesired paths in $\mathcal{G}$, e.g. removing the influence of $U$ on $Y$ and/or $Z$ in Figure 1(a), leading to a new graph $\mathcal{G}^0$. While this interpretation is correct for joint balancing in the case of Figure 1(a), Proposition 5.1 below (proof in Appendix C) shows that it can be erroneous in general: the distribution $Q$ underlying the balanced data might not factorize according to $\mathcal{G}^0$ and therefore might not obey the statistical dependence relationships implied by $\mathcal{G}^0$. Therefore, balancing data to make $Z$ and $Y$ statistically independent, i.e. selecting samples in proportion to $P^t(Z)P^t(Y)/P^t(Z,Y)$, is not equivalent to generating data from a distribution that factorises according to $\mathcal{G}^0$ in general. This factorization is important because downstream distributions $P'(X,Y,Z)$ are often assumed to follow this factorization; in fact, this assumption underlies a number recommendations for applying regularization methodologies such as in [74].

**Proposition 5.1.** *Let $\langle \mathcal{G}, P^t \rangle$ be the CBN underlying the data, where $\mathcal{G}$ contains an undesired path between $Z$ and $Y$, and let $\mathcal{G}^0$ be a modification of $\mathcal{G}$ in which the undesired path has been removed. The distribution $Q$ obtained by jointly balancing the data need not factorize according to $\mathcal{G}^0$.*

Proposition 5.1 shows that statistical (in)dependencies that we assumed would remain fixed (i.e. the black edges on the graph) can be modified by the process of joint balancing. As a consequence, further interventions on $Q$ (e.g. the addition of a regularizer) should not be motivated by $\mathcal{G}^0$, and we show below that combining data balancing with other mitigation strategies can lead to unexpected results.

### 5.1 Data balancing can hinder regularization and vice-versa

When confronted with a failure mode, it is reasonable to ask whether an additional fairness or robustness regularizer on the training loss might be beneficial. Based on Proposition 5.1, we see that this question might have a different answer if we are in $P^t$ or in $Q$. Below, we consider each failure mode and ask whether performing an additional regularization motivated by the literature would mitigate the undesired dependencies in $Q$. The results are summarized in Table 1, with suggested next steps. In Appendix C.1.2, we discuss when balancing with regularization is sufficient for different fairness criteria.

**Anti-causal task.** In the case of an anti-causal task with a dependence between $Y$ and $Z$ (Figures 1(a,c,d)), Veitch et al. [74] recommend to impose an independence between $f(X)$ and $Z$ conditioned

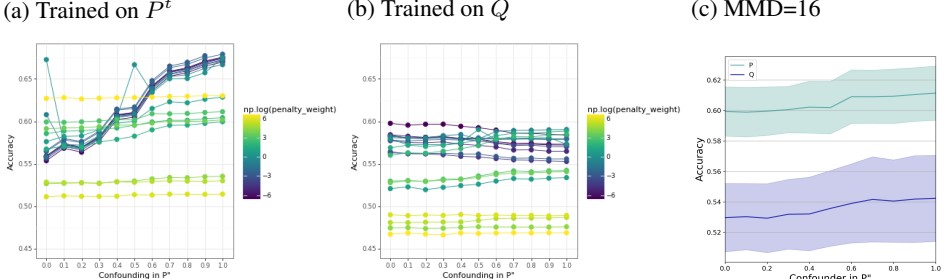

Figure 3: Accuracy across different values of the confounder strength (i.e. different $P' \in \mathcal{P}$), for each value of MMD regularization considered (displayed by the color gradient). (a) Models trained on $P^t$. (b) Models trained on $Q$. Results are averaged across seeds for clarity. Notice the different y-scales. (c) Displays the mean and standard deviation across seeds for MMD=16.

on $Y$. If we consider both the purely spurious correlation and the entangled case, we see that regularization and data balancing would have the same effects of blocking any dependence between $\{Y, X_{\bar{Z}}^{\perp}\}$ and $\{Z, X_Y^{\perp}, X_{Y \wedge Z}\}$. We demonstrate that $X_{\bar{Z}}^{\perp} \perp\!\!\!\perp Z \mid Y$ in both $P^t$ and $Q$ (see Appendix C.1), and this regularization is sensible under both distributions. This means that performing the regularization provides the sufficient conditions for a risk-invariant model, whether or not joint data balancing is performed. In theory, data balancing is not needed but is also not harmful. In the case of an added confounder, we have that $X_V$ depends on both $Y$ and $Z$ due to non-causal paths through $V$. Therefore, imposing that $f(X) \perp\!\!\!\perp_Q Z \mid Y$ might lead to results whereby the model only depends on $V$ or is trivial (e.g. predicts a constant) as the regularization encourages the removal of any dependence on $Z$, which is related to $Y$ via $X_V$. This behavior would be observed in both $P^t$ and $Q$, but data balancing on its own might be less detrimental than regularization in terms of predictive power even though it does not resolve all undesired dependencies. In this case, regularization hinders data balancing.

Based on the balanced data from Section 3, we add a conditional Maximum Mean Discrepancy [MMD, 28] to encourage $f(X) \perp\!\!\!\perp_Q Z \mid Y$ during training, varying the strength of this regularizer via a hyper-parameter. In the case of the purely spurious statistical dependence between $Y$ and $Z$ (Figure 1(a)), there is little variation between the metrics across MMD strengths, and the model is fair and robust (Figure 2(left)). In the entangled case (Figure 2(right)), the model's performance on $Q$ and $P^0$ are close for medium values of the hyper-parameter (before MMD overpowers the training) and worst group performance improves markedly. This result suggests that, with the added regularizer, $f$ only varies with $X_{\bar{Z}}^{\perp}$). Performing the same regularization in the presence of another confounder (Figure 2(middle)) leads to a plateau in performance on $Q$, but low performance on $P^0$ and chance-level worst group performance. In this case, we posit that the model relies exclusively on $X_V$ for its predictions, and the regularizer is detrimental compared to data balancing on its own (MMD=0 on the plot).

**Causal task.** Finally, let us consider the causal task in Figure 1(b). In a similar case, Veitch et al. [74] suggests a regularizer such that $f(X) \perp\!\!\!\perp_{P^t} Z$, which would encourage the model $f(X)$ to vary only with $X_{\bar{Z}}^{\perp}$ as $X_{\bar{Z}}^{\perp} \perp\!\!\!\perp_{P^t} Z$. However, data balancing induces a dependence between $X_{\bar{Z}}^{\perp}$ and $Z$, as expressed below:

$$Q(X_{\bar{Z}}^{\perp} \mid Z) = \frac{\sum_{X_Y^{\perp}, Y} P^t(X_{\bar{Z}}^{\perp}, X_Y^{\perp} \mid Z, Y) P^t(Z) P^t(Y)}{\sum_{X_Y^{\perp}, X_{\bar{Z}}^{\perp}, Y} P^t(X_{\bar{Z}}^{\perp}, X_Y^{\perp} \mid Z, Y) P^t(Z) P^t(Y)} = \sum_Y P^t(X_{\bar{Z}}^{\perp} \mid Z, Y) P^t(Y),$$

The RHS cannot be simplified further because $X_{\bar{Z}}^{\perp} \not\perp\!\!\!\perp_{P^t} Z \mid Y$, because $Y$ is a collider under $P^t$. Thus, the left hand side is a function of $Z$ in general (see Appendix C.1 for further details and a numerical simulation). In this case, regularizing to enforce $f(X) \perp\!\!\!\perp_Q Z$ would destroy information in $X_{\bar{Z}}^{\perp}$, whereas the same regularization under $P^t$ would have enabled $f(X)$ to use all of the information in $X_{\bar{Z}}^{\perp}$. Therefore, data balancing may hinder regularization.

We illustrate this result on the Amazon reviews dataset from Section 3 by imposing a marginal MMD regularization $f(X) \perp\!\!\!\perp Z$ during training and evaluating risk-invariance across multiple $P' \in \mathcal{P}$. When training on $P^t$, we observe that the regularization allows to 'flatten' the curve, such that from

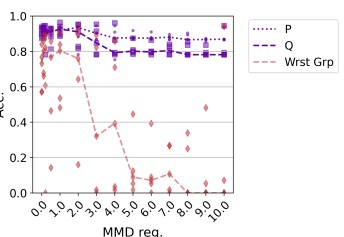

Table 3: VGG model performance on CelebA, when trained on $P^t$, on $Q$, with ImageNet pre-training ('Pre-trained') on $Q$, with MMD ('MMD') on $P^t$ with regularizer=5. All models are evaluated on $Q$.

| Model | Acc. ($\uparrow$) | Worst Grp ($\uparrow$) | Encoding ($\sim 0.5$) | Equ. Odds ($\downarrow$) |
|---|---|---|---|---|
| ERM on $P^t$ | $0.791 \pm 0.037$ | $0.314 \pm 0.093$ | $0.868 \pm 0.015$ | $0.243 \pm 0.036$ |
| ERM on $Q$ | $0.839 \pm 0.022$ | $0.674 \pm 0.088$ | $0.709 \pm 0.066$ | $0.125 \pm 0.022$ |
| Pre-trained on $Q$ | $0.874 \pm 0.006$ | $0.726 \pm 0.037$ | $0.740 \pm 0.033$ | $0.111 \pm 0.010$ |
| MMD on $P^t$ | $0.813 \pm 0.036$ | $0.146 \pm 0.172$ | $0.630 \pm 0.010$ | $0.001 \pm 0.002$ |

Figure 4: Model performance on test sets sampled from $P^t$ (dotted) and $Q$ (dashed). The model is trained on $P^t$ with regularization $f(X) \perp\!\!\!\perp Z \mid Y$.

medium to high values of MMD regularization, the model is risk-invariant (Figure 3(a)). On the jointly balanced data, medium values of the regularization degrade risk-invariance (see green curves on Figure 3(b)). Overall, model performance is also lower for the models trained on $Q$ compared to models trained on $P^t$ across test sets from $P' \in \mathcal{P}$, at similar levels of regularization (see Figure 3(c) for MMD=16). This result displays that $X_{\bar{Z}}^{\perp}$ is not a sufficient statistic for $Y$ in $Q$.

## 6 Case study: distinguishing between failure modes in CelebA

In this section, we show that when $Y$ and $Z$ are available at training time, we can try to distinguish between failure modes of data balancing by using our different observations, even in the absence of a full causal graph. We illustrate this using the benchmark task of detecting blond hair in pictures of celebrities in the CelebA [46] dataset. This label has a strong correlation with perceived gender: half of the non-males have blond hair, while only $\sim 7\%$ of males do. We consider a balanced, subsampled dataset (train: $n = 4,096$, test/valid: $n = 400$) and the original, confounded dataset. We train a VGG [68] and four Vision Transformer [ViT, 18] architectures, with number of parameters ranging from 17 to 690 millions.

We observe that, while training with balanced data leads to higher worst group accuracy and lower equalized odds scores than training with the historical data (Table 3), an important gap remains between the overall and worst group performances. These results show that data balancing leads to improvements in downstream fairness and robustness metrics, but does not provide a risk-invariant or fair model on its own. Therefore, it is likely that one of the conditions for data balancing to be sufficient is not fulfilled and understanding which condition is violated can guide our selection of another technique.

**Distinguishing between failure modes.** We first assume that the task is anti-causal. We then aim to understand whether there is another confounder, the data is entangled, or the representation is entangled (Proposition 4.4). As per Kirichenko et al. [41], we first attempt to improve our representation by pre-training the VGG with ImageNet [16]. While we observe an increase in performance with pre-training, there is no clear decrease in equalized odds. This result suggests that the failure may lie elsewhere. We then train models with MMD on $P^t$, with the expectation that we would observe a plateau for entangled data when the model learns $f(X_{\bar{Z}}^{\perp})$, or a stark decrease in worst group performance in the presence of another confounder. While there is no major pattern of correlation between $Y$ and another attribute in the balanced data (see Appendix E.2.2), small effects might combine, or there might be other, unobserved attributes that influence $Y$. For a medium value of the regularization hyper-parameter, the model displays a plateau in performance and poor worst group performance. This result suggests an effect of another confounder and next steps can include methods such as Alabdulmohsin et al. [2], which controls for all (observed) auxiliary factors of variation.

## 7 Related works

**Balanced data as mitigation for invariant models.** Our results extend those of Makar et al. [48] which considered a single causal graph. Wang et al. [76] displayed that balancing data did not lead to

a reduction in bias amplification. The authors posit that this failure of balanced data to correct for spurious signals is due to unobserved confounding factors which is confirmed in Alabdulmohsin et al. [2]. Rolf et al. [63] investigated upsampling by relying on a scaling law per group, focusing on the question of fairness vs performance trade-off [22]. Focusing on causal NLP settings, Joshi et al. [36] investigated causal and non-causal features, concluding that data balancing does not help in all cases. Closer to our work is that of Puli et al. [58], in which the authors showed that having $Y \perp\!\!\!\perp_Q Z$ does not imply that $Y \perp\!\!\!\perp_Q Z \,|\, X$ and the model can learn signals related to $Z$. Puli et al. [58] propose a method to learn a representation $r$ such that $Y \perp\!\!\!\perp Z \,|\, r(X)$. Our work provides a framework to understand these different failure modes and proposes strategies to distinguish between them. While we focus on pre-processing mitigation with a fixed distribution $Q(X, Y, Z)$, another line of work considers dynamic resampling in-processing [e.g. 35, 61, 12]. As the resampling converges towards a fixed distribution $P'(Z|Y)$, we would expect failure modes in the presence of entangled data or of another confounder. Nevertheless, the variation in $P'(Z|Y)$ at the early stages of training might be beneficial, e.g. by disentangling the representation. We leave this investigation for future work.

**Causal feature selection.** Some works have used a causal framing to select features such that $f(X)$ has robustness and/or fairness properties [e.g. 47, 71, 69, 25, 67, 38]. Similarly, our work defines independence conditions on covariates to obtain an optimal, invariant model, and can be used to select features. Two major distinctions between feature selection works and ours reside in the fact that we consider the case in which we do not observe $X_Z^\perp$ explicitly and that we investigate the impact of data balancing.

# 8 Discussion

In this work, we uncover important results to guide the use of data balancing for mitigating undesired dependencies between covariates, outcomes and auxiliary factors of variation. We first show (Section 3) that joint data balancing might not achieve the desired fairness or robustness criteria, and that the failures may seem difficult to predict. Motivated by these results, we introduce conditions under which data balancing leads to a robust or fair model (Sections 4, B.2). Importantly, we show that data balancing is not equivalent to 'dropping an edge' in the causal graph and can lead to distributions that do not factorize according to the desired graph (Section 5). This can have downstream consequences if further mitigation strategies are motivated by the causal graph and highlights why regularization and data balancing might not go 'hand in hand'. This last result shows that data balancing should not be performed as a 'default', and mitigation strategies should be based on the causal graph of the application. Finally, even in the absence of a causal graph, our findings may help to pinpoint which condition(s) are not fulfilled, and guide further mitigation (Section 6).

**Limitations.** The conditions defined in Section 4 for risk-invariance depend on the expression of $\mathcal{P}$ as a correlation shift [48, 62]. Other expressions or shifts are likely to lead to other conditions. In our experiments, we have mostly subsampled datasets to obtain balanced distributions. We would expect similar results for other joint balancing methods. Variations are, however, possible due to the finite-set nature of the computations [48], e.g. with reweighting displaying more variance [33], potentially under-performing in overparametrized settings [11, 65]. We also note that, while we aimed to provide upper bounds for the effectiveness of data balancing, we did not use additional training strategies for mitigation beyond regularization. We believe that our causal framework can be a useful tool to analyze other pre- or in-processing methods that enforce independence between variables in the data generating process [e.g. 1, 58]. On the other hand, our framework might not be suited to analyze the effects of other mitigation strategies, e.g. hyper-parameter optimization [57]. We discuss the broader societal impacts of our work in Appendix E.2.2.

**Future work.** This work considered a variety of causal graphs in order to provide general insights rather than task-specific conditions. However, investigating specific graphs could enable to leverage further strategies including other balancing techniques [e.g. 38, 72]. We believe that our causal framing could then be a useful resource to analyze the effect of these strategies on downstream fairness and robustness criteria. Finally, we illustrate our propositions with binary classification tasks and confounders. While our reasoning applies to more complex settings, there might be further considerations to account for when generalizing beyond binary variables, especially with respect to estimation.

## Acknowledgments and Disclosure of Funding

We thank Virginia Aglietti for feedback on this work and Victor Veitch for sharing experimental code for the Amazon reviews experiments. This work was funded by Google DeepMind.

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

# A Failure modes of data balancing

## A.1 Failure mode: Balancing on one variable can increase bias

It is common to consider balancing on classes or groups as it requires fewer labels than joint balancing. However, without further intervention, class or group balancing on its own does not provide an invariant model when $Y$ and $Z$ are marginally dependent [e.g. 44]. In Figure 1(a), this means that $X \overset{\perp}{Z} \not\perp_Q Z \mid Y$, invalidating Prop.4.2. Below, we formalize the observation in Yan et al. [78] that balancing on one variable might affect the representation of the other, and provide bounds on the impact of this strategy.

**Formalization and proof.**  We formalize this issue in Proposition A.1 for the binary case with a binary attribute.

**Proposition A.1.** *Consider data balancing of $Y$; the marginal of $Z$ will be farther from uniform than the marginal of $Z$ before balancing if*

$$\mathrm{sgn}\left(\frac{P^t(Z=1) - \frac{1}{2}}{P^t(Y=1) - \frac{1}{2}}\right) = \mathrm{sgn}\left(\mathbb{E}[Z|Y=0] - \mathbb{E}[Z|Y=1]\right).$$

Intuitively, if the biases of $Y$ and $Z$ are in the same (resp. opposite) direction, then this condition is satisfied if $Z$ has a negative (resp. positive) correlation with $Y$. For example, if we have $P^t(Y=1) = \frac{1}{4}$, $\mathbb{E}[Z|Y=1] = 1$ and $\mathbb{E}[Z|Y=0] = \frac{1}{3}$, then $\mathbb{E}[Z] = \frac{1}{2}$ before balancing but $\mathbb{E}[Z] = \frac{1}{3}$ after balancing.

*Proof of Proposition A.1.* We assume that $Y$ and $Z$, representing the label and confounder, are both binary. We will data-balance on $Y$. Let $Z \mid S$ denote the distribution of $Z$ after data balancing. To characterize when the distribution of $Z \mid S$ is farther from uniform than the distribution of $Z$, we will first derive

$$\mathbb{E}[Z] - \frac{1}{2} = P^t(Y=1)\left(\mathbb{E}[Z \mid Y=1] - \frac{1}{2}\right) + P^t(Y=0)\left(\mathbb{E}[Z \mid Y=0] - \frac{1}{2}\right)$$

and

$$\mathbb{E}[Z \mid S] - \frac{1}{2} = \frac{1}{2}\left(\mathbb{E}[Z \mid Y=1] - \frac{1}{2}\right) + \frac{1}{2}\left(\mathbb{E}[Z \mid Y=0] - \frac{1}{2}\right).$$

Now, taking the difference, we have

$$\mathbb{E}[Z] - \frac{1}{2} = \mathbb{E}[Z \mid S] - \frac{1}{2} + \left(P^t(Y=1) - \frac{1}{2}\right)\left(\mathbb{E}[Z \mid Y=1] - \frac{1}{2}\right) + \left(P^t(Y=0) - \frac{1}{2}\right)\left(\mathbb{E}[Z \mid Y=0] - \frac{1}{2}\right)$$

$$= \mathbb{E}[Z \mid S] - \frac{1}{2} + \left(P^t(Y=1) - \frac{1}{2}\right)\mathbb{E}[Z \mid Y=1] + \left(P^t(Y=0) - \frac{1}{2}\right)\mathbb{E}[Z \mid Y=0]$$

$$= \mathbb{E}[Z \mid S] - \frac{1}{2} + \left(P^t(Y=1) - \frac{1}{2}\right)\left(\mathbb{E}[Z \mid Y=1] - \mathbb{E}[Z \mid Y=0]\right).$$

We can derive some sufficient conditions for bias increase, which occurs when $|\mathbb{E}[Z \mid S] - \frac{1}{2}| \geq |\mathbb{E}[Z] - \frac{1}{2}|$. We proceed by cases. If $\mathbb{E}[Z] - \frac{1}{2} > 0$, then

$$\mathbb{E}[Z \mid S] - \frac{1}{2} = \mathbb{E}[Z] - \frac{1}{2} + \left(P^t(Y=1) - \frac{1}{2}\right)\left(\mathbb{E}[Z \mid Y=1] - \mathbb{E}[Z \mid Y=0]\right)$$

$$= \left|\mathbb{E}[Z] - \frac{1}{2}\right| + \left(P^t(Y=1) - \frac{1}{2}\right)\left(\mathbb{E}[Z \mid Y=1] - \mathbb{E}[Z \mid Y=0]\right),$$

so $\left|\mathbb{E}[Z \mid S] - \frac{1}{2}\right| = \left|\mathbb{E}[Z] - \frac{1}{2}\right| + \left(P^t(Y=1) - \frac{1}{2}\right)\left(\mathbb{E}[Z \mid Y=1] - \mathbb{E}[Z \mid Y=0]\right)$. Thus, the bias gets worse if $\left(P^t(Y=1) - \frac{1}{2}\right)\left(\mathbb{E}[Z \mid Y=1] - \mathbb{E}[Z \mid Y=0]\right) > 0$.

Similar reasoning shows that if $\mathbb{E}[Z] - \frac{1}{2} < 0$, then

$$\left|\mathbb{E}[Z \mid S] - \frac{1}{2}\right| = \left|\mathbb{E}[Z] - \frac{1}{2}\right| - \left(P^t(Y=1) - \frac{1}{2}\right)\left(\mathbb{E}[Z \mid Y=1] - \mathbb{E}[Z \mid Y=0]\right),$$

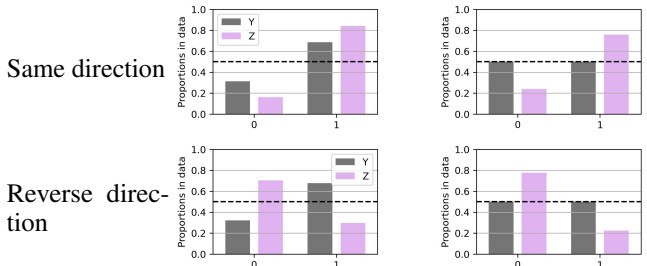

Figure 5: Proportions of $Y = 0, 1$ (grey bars) and $Z = 0, 1$ (purple bars) before (left) and after (right) balancing the data on $Y$.

and we can conclude that the bias is worsened if $\left(P^t(Y = 1) - \frac{1}{2}\right)\left(\mathbb{E}[Z \mid Y = 1] - \mathbb{E}[Z \mid Y = 0]\right) < 0$. Taking both statements together, we obtain the statement of the proposition. $\square$

For example, if we have $P^t(Y = 1) = \frac{1}{4}$, $\mathbb{E}[Z \mid Y = 1] = 1$ and $\mathbb{E}[Z \mid Y = 0] = \frac{1}{3}$, then $\mathbb{E}[Z] = \frac{1}{2}$ but $\mathbb{E}[Z \mid S] - \frac{1}{2} = \frac{1}{6}$; despite $Z$ starting as unbiased, the data balancing induces a bias of $\frac{1}{6}$.

There are a few implications of this derivation. First, we obtain an easy upper bound for the worsening of the bias of $Z$ caused by data balancing: taking absolute values of both sizes and using the triangle inequality on the right yields

$$\left|\mathbb{E}[Z] - \frac{1}{2}\right| \le \left|\mathbb{E}[Z \mid S] - \frac{1}{2}\right| + \left|P^t(Y = 1) - \frac{1}{2}\right| \left|\mathbb{E}[Z \mid Y = 1] - \mathbb{E}[Z \mid Y = 0]\right|,$$

Bringing the second term over to the left hand side and applying the same logic produces

$$\left|\mathbb{E}[Z \mid S] - \frac{1}{2}\right| \le \left|\mathbb{E}[Z] - \frac{1}{2}\right| + \left|P^t(Y = 1) - \frac{1}{2}\right| \left|\mathbb{E}[Z \mid Y = 1] - \mathbb{E}[Z \mid Y = 0]\right|,$$

and combining both terms shows that the difference in bias of $Z$ and $Z \mid S$ is bounded by

$$\left|\left|\mathbb{E}[Z] - \frac{1}{2}\right| - \left|\mathbb{E}[Z \mid S] - \frac{1}{2}\right|\right| \le \left|P^t(Y = 1) - \frac{1}{2}\right| \left|\mathbb{E}[Z \mid Y = 1] - \mathbb{E}[Z \mid Y = 0]\right|.$$

**Simulation.** We present a simple simulation to illustrate our reasoning: $U \sim \mathcal{N}(0, 0.1)$ is a common cause to $Z$ and $Y$. More specifically, the continuous distributions of $Y$ and $Z$ both have the form $U + \epsilon$, with $\epsilon \sim \mathcal{N}(0.05, 0.02)$. We then binarize $Y$ by thresholding at 0. This creates an imbalance in the marginal of $Y$, such that a random sample of 5,000 examples has $\sim 68\%$ of positive labels. We then want to vary the marginal of $Z$, which also requires affecting their correlation. To this end, we vary the threshold for binarizing $Z$. This leads us to 2 main cases: for thresholds above 0 (i.e. $Y$'s threshold), the marginal of $Z$ is imbalanced in the same direction as that of $Y$. For thresholds smaller than 0., we obtain the opposite, i.e. if $Y = 1$ is over-represented, $Z = 1$ is under-represented.

We illustrate these 2 cases in Figure 5. We observe that when the marginals are similar, balancing $Y$ brings $Z$ closer to a uniform distribution (top row). However, the marginal distribution of $Z$ becomes more imbalanced after balancing on $Y$ if the two distributions are reversed (bottom row). When the correlation is small, there is little change in the marginal of $Z$ when balancing on $Y$, which is expected.

For completeness, we perform 200 simulations with different thresholdings for $Z$ and present the results in Figure 6.

## A.2 Failure mode: entangled signals

In the case where $X$ includes non-trivial intersection information $X_{Y \wedge Z}$, data balancing will in general be insufficient to ensure that there is no association bias. This is because a risk-minimizing predictor $f(X)$ will condition on $X_{Y \wedge Z}$, and the distribution of these intersection features is influenced by $Z$.

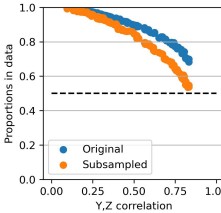 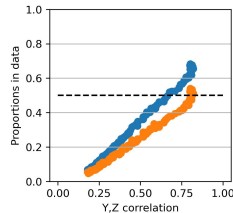

Figure 6: Distribution $P^t(Z = 1)$ before (blue) and after (orange) balancing the data according to $Y$, for different values of the binarization threshold of $Z$ which translates into different correlation coefficients between $Y$ and $Z$. Left: similar direction of under-representation. Right: opposite direction.

Specifically, we will give a case where $Y$ is marginally independent of $Z$ and there is no uncontrolled confounding, but $E[f(X) \mid Z = 0] \neq E[f(X) \mid Z = 1]$.

Suppose we have the following data generating process (DGP):

$$P^t(Y = 1) = 0.5$$
$$P^t(Z = 1) = 0.5$$
$$P^t(Y = 1, Z = 1) = P^t(Y = 1)P^t(Z = 1), \text{i.e., } Y_Z^{\perp}$$
$$P^t(X = 1) = \begin{cases} p & \text{if } Y \text{ OR } Z \\ q & \text{o.w.} \end{cases}$$

Note that in this case the entirety of $X$ would be classified as intersection information $X_{Y \wedge Z}$.

In this setup, the Bayes-optimal probabilities for classification, $f(X)$, are given by:

$$f(1) := P^t(Y = 1 \mid X = 1) = \frac{P^t(X = 1 \mid Y = 1)P^t(Y = 1)}{P^t(X = 1)} = \frac{p \cdot 0.5}{0.75p + 0.25q}$$

and

$$f(0) := P^t(Y = 1 \mid X = 0) = \frac{(1 - P^t(X = 1 \mid Y = 1))P^t(Y = 1)}{P^t(X = 0)} = \frac{(1 - p) \cdot 0.5}{1 - (0.75p + 0.25q)}$$

Note that when we condition on $Z = 0, 1$, the expectation of $f(X)$ is different whenever (1) $p \neq q$, i.e., whenever the distribution of $X$ actually depends on the function of $Y$ and $Z$, and (2) $f(1) \neq f(0)$, i.e., there is some information in $X$ to predict $Y$:

$$E[f(X) \mid Z = 1] = E[E[f(X) \mid X, Z = 1]] = pf(1) + (1 - p)f(0)$$

$$E[f(X) \mid Z = 0] = E[E[f(X) \mid X, Z = 0]] \\ = (0.5p + 0.5q)f(1) + (0.5(1 - p) + 0.5(1 - q))f(0)$$

In the simple case where $p = 1$ and $q = 0$ (i.e., $X = Y$ OR $Z$ deterministically), we get

$$f(X) := P^t(Y = 1 \mid X) = \begin{cases} 2/3 & \text{if } X = 1 \\ 0 & \text{if } X = 0. \end{cases}$$

$$E[f(X) \mid Z] = \begin{cases} 2/3 & \text{if } Z = 1 \\ 1/3 & \text{if } Z = 0. \end{cases}$$

## B  Conditions for data balancing to lead to an invariant and optimal model

We first investigate the case of a risk-invariant model w.r.t $\mathcal{P}$, and then discuss fairness criteria.

## B.1 Risk-invariant, optimal model

In this section we provide proofs for Section 4.

Recall that $\mathcal{P} = \{P'(X,Y,Z) = P^t(X_Z^{\perp}|Y,Z)P^t(X_Y^{\perp}|Y,Z)P^t(X_{Z \wedge Y}|Y,Z)P'(Z|Y)P^t(Y)\}$ and that we assume a data balancing distribution $Q(X,Y,Z) \in \mathcal{P}$ of the form $Q(X,Y,Z) = P^t(X \mid Y,Z)P^t(Z)P^t(Y)$. Also recall that we define $X_Z^{\perp}$ to be a sufficient statistic for $Y$ in $Q$ if $\mathbb{E}_Q[Y \mid X] = \mathbb{E}_Q[Y \mid X_Z^{\perp}]$.

**Proposition 4.2.** *If $X_Z^{\perp} \perp\!\!\!\perp_Q Z \mid Y$ and $X_Z^{\perp}$ is a sufficient statistic for $Y$ in $Q$, then the risk-minimizer $f(X) := \mathbb{E}_Q[Y \mid X]$ is risk-invariant and optimal w.r.t. $\mathcal{P}$.*

*Proof.* Let $P'$ be an arbitrary distribution in $\mathcal{P}$. We have

$$
\begin{aligned}
P'(X_Z^{\perp} \mid Y) &= \sum_Z P'(X_Z^{\perp} \mid Y,Z)P'(Z \mid Y) \\
&\overset{(1)}{=} \sum_Z Q(X_Z^{\perp} \mid Y, \cancel{Z})P'(Z \mid Y) \\
&= Q(X_Z^{\perp} \mid Y),
\end{aligned}
$$

where (1) holds as $P',Q \in \mathcal{P}$ and by the independence assumption. As $P'(Y) = Q(Y)$ we obtain $P'(X_Z^{\perp},Y) = Q(X_Z^{\perp},Y)$. As $X_Z^{\perp}$ is a sufficient statistic for $Y$ in $Q$, $f(X) := \mathbb{E}_Q[Y \mid X] = \mathbb{E}_Q[Y \mid X_Z^{\perp}]$, that is $f(X)$ (and therefore the loss $\ell(f;X,Y)$) remains constant for different values of $X_Y^{\perp}, X_{Y \wedge Z}$, giving

$$
\mathbb{E}_{X,Y \sim P'}[\ell(f;X,Y)] = \mathbb{E}_{X_Z^{\perp},Y \sim P'}[\ell(f;X,Y)] = \mathbb{E}_{X_Z^{\perp},Y \sim Q}[\ell(f;X,Y)].
$$

The same reasoning can be repeated for $P'' \in \mathcal{P}$, obtaining $\mathbb{E}_{X,Y \sim P'}[\ell(f;X,Y)] = \mathbb{E}_{X,Y \sim P''}[\ell(f;X,Y)]$, which proves that $f$ is risk-invariant w.r.t. $\mathcal{P}$.
As $f = \min_{f'} \mathbb{E}_{X,Y \sim Q}[\ell(f';X,Y)]$ and $\mathbb{E}_{X,Y \sim P'}[\ell(f;X,Y)] = \mathbb{E}_{X,Y \sim Q}[\ell(f;X,Y) \, \forall P' \in \mathcal{P}$, we obtain $f = \min_{f'} \mathbb{E}_{X,Y \sim P'}[\ell(f';X,Y)]), \forall P' \in \mathcal{P}$, which implies that $f$ is optimal w.r.t. $\mathcal{P}$. $\quad\square$

**Corollary 4.3.** *Let $R = \{X_Y^{\perp}, X_{Y \wedge Z}\}$. If $R \perp\!\!\!\perp_{P^t} \{X_Z^{\perp},Y\} \mid Z$ and $X_Z^{\perp} \perp\!\!\!\perp_{P^t} Z \mid Y$, then $f(X_Z^{\perp}) = \mathbb{E}_Q[Y \mid X_Z^{\perp}]$ is risk-invariant and optimal w.r.t. $\mathcal{P}$.*

*Proof.* We have

$$
\begin{aligned}
Q(Y \mid R, X_Z^{\perp}) &= \frac{\sum_Z Q(R, X_Z^{\perp},Y,Z)}{\sum_{Z,Y} Q(R, X_Z^{\perp},Y,Z)} \\
&\overset{(1)}{=} \frac{\sum_Z P^t(R, X_Z^{\perp} \mid Y,Z)P^t(Z)P^t(Y)}{\sum_{Z,Y} P^t(R, X_Z^{\perp} \mid Y,Z)P^t(Z)P^t(Y)} \\
&\overset{(2)}{=} \frac{\sum_Z P^t(R \mid \cancel{X_Z^{\perp},Y},Z)P^t(X_Z^{\perp} \mid Y,\cancel{Z})P^t(Z)P^t(Y)}{\sum_{Z,Y} P^t(R \mid \cancel{X_Z^{\perp},Y},Z)P^t(X_Z^{\perp} \mid Y,\cancel{Z})P^t(Z)P^t(Y)} \\
&= \frac{P^t(R)P^t(X_Z^{\perp} \mid Y)P^t(Y)}{P^t(R)\sum_Y P^t(X_Z^{\perp} \mid Y)P^t(Y)} \\
&= P^t(Y \mid X_Z^{\perp}),
\end{aligned}
$$

where (1) holds by the definition of the balanced distribution $Q$ and (2) holds by the independence assumptions. This derivation shows that $Y \perp\!\!\!\perp_Q R \mid X_Z^{\perp}$ and therefore that $X_Z^{\perp}$ is a sufficient statistic for $Y$ in $P^t$. We are in the same conditions as in Proposition 4.2, which implies that $f$ is risk-invariant and optimal w.r.t. $\mathcal{P}$. $\quad\square$

**Proposition 4.4.** *Let $\hat{f} \in \mathcal{F}$ be some fitted model and $\epsilon > 0$. Assume that, for all $P',P'' \in \mathcal{P}$, we have $\left| \mathbb{E}_{P'}[Y \mid \hat{f}(X,Y)] - \mathbb{E}_{P'}[Y \mid X_Z^{\perp}] \right| \leq \frac{\epsilon}{2}$. Then $\hat{f}$ is $\epsilon$-risk invariant, meaning that*

$$
\sup_{P',P'' \in \mathcal{P}} R_{P'}(\hat{f}) - R_{P''}(\hat{f}) \leq \epsilon.
$$

*Proof.* By definition, $f^*$ is risk-invariant w.r.t. $\mathcal{P}$ and optimal. By the triangle inequality, we can then write

$$\begin{aligned}
|R_{P'}(\hat{f}) - R_{P''}(\hat{f})| &\leq |R_{P'}(\hat{f}) - R_{P'}(f^*)| + |R_{P'}(f^*) - R_{P''}(\hat{f})| \\
&\leq |R_{P'}(\hat{f}) - R_{P'}(f^*)| + |R_{P''}(f^*) - R_{P''}(\hat{f})| \\
&\leq \frac{\epsilon}{2} + \frac{\epsilon}{2}.
\end{aligned}$$

$\square$

## B.2 Conditions for data balancing to lead to a fair model

In this section, we focus on fairness definitions and provide sufficient conditions for data balancing to lead to a fair model. The results we describe do not address the case where $X_Z^{\perp}$ is not accessible directly.

**Proposition B.1** (Demographic parity). $X_Z^{\perp} \perp\!\!\!\perp_Q Z$ if $X_Z^{\perp} \perp\!\!\!\perp_{P^t} Z \,|\, Y$; that is balancing successfully induces independence between $X_Z^{\perp}$ and $Z$ if $X_Z^{\perp}$ and $Z$ are independent given $Y$ in the original data distribution.

*Proof.* First, note that, under the assumed conditional independences,

$$\begin{aligned}
Q(X_Z^{\perp}, Y, Z) &= \sum_{X_Y^{\perp}, X_{Y \wedge Z}} Q(X_Z^{\perp}, X_Y^{\perp}, X_{Y \wedge Z}, Y, Z) \\
&= \sum_{X_Y^{\perp}, X_{Y \wedge Z}} P^t(X_Z^{\perp}, X_Y^{\perp}, X_{Y \wedge Z} | Y, Z) P^t(Y) P^t(Z) \\
&= P^t(X_Z^{\perp} | Y, Z) P^t(Y) P^t(Z) \\
&= P^t(X_Z^{\perp} | Y) P^t(Y) P^t(Z),
\end{aligned}$$

where the first equality is by applying the definition of $Q$ and the last line follows from the assumed invariance. We can then derive

$$\begin{aligned}
Q(X_Z^{\perp} \,|\, Z) &= \frac{\sum_Y Q(X_Z^{\perp}, Y, Z)}{\sum_{Y, X_Z^{\perp}} Q(X_Z^{\perp}, Y, Z)} \\
&= \frac{\sum_Y P^t(X_Z^{\perp} | Y) P^t(Y) P^t(Z),}{\sum_{Y, X_Z^{\perp}} P^t(X_Z^{\perp} | Y) P^t(Y) P^t(Z)} \\
&= \sum_Y P^t(X_Z^{\perp} \,|\, Z, Y) P^t(Y) \\
&= \sum_Y P^t(X_Z^{\perp} \,|\, Y) P^t(Y) \\
&= P^t(X_Z^{\perp}).
\end{aligned}$$

This equality implies that $Q(X_Z^{\perp} \,|\, Z)$ does not depend on $Z$, verifying the conditional independence claim. $\square$

*Remark* B.2. We would not expect $X_Z^{\perp}$ and $Z$ to be independent in $Q$ if $X_Z^{\perp}$ and $Z$ are not independent given $Y$ in $P^t$. In the previous proof, we derived

$$Q(X_Z^{\perp} | Z) = \sum_Y P^t(X_Z^{\perp} \,|\, Z, Y) P^t(Y).$$

Except for some corner cases, we would expect that $P^t(X_Z^{\perp} \,|\, Z, Y)$ would not vary with $Z$ if $Q(X_Z^{\perp} | Z)$ does not.

**Proposition B.3** (Predictive parity). $Y \perp\!\!\!\perp_Q Z \,|\, X_Z^{\perp}$ if $X_Z^{\perp} \perp\!\!\!\perp_{P^t} Z \,|\, Y$; that is, data balancing successfully induces independence between $Y$ and $Z$ given $X_Z^{\perp}$ if $X_Z^{\perp}$ and $Z$ are independent given $Y$ in the original data distribution.

*Proof.* Let $X_{\overline{Z}}^{\perp} \perp\!\!\!\perp_{P^t} Z \mid Y$. The following derivation demonstrates the claim,

$$
\begin{aligned}
Q(Y \mid X_{\overline{Z}}^{\perp}, Z) &= \frac{Q(X_{\overline{Z}}^{\perp}, Y, Z)}{\sum_Y Q(X_{\overline{Z}}^{\perp}, Y, Z)} \\
&\overset{(1)}{=} \frac{P^t(X_{\overline{Z}}^{\perp} \mid Y, Z) P^t(Y) P^t(Z)}{\sum_Y P^t(X_{\overline{Z}}^{\perp} \mid Y, Z) P^t(Y) P^t(Z)} \\
&\overset{(2)}{=} \frac{P^t(X_{\overline{Z}}^{\perp} \mid Y) P^t(Y) P^t(Z)}{\sum_Y P^t(X_{\overline{Z}}^{\perp} \mid Y) P^t(Y) P^t(Z)} \\
&= P^t(Y \mid X_{\overline{Z}}^{\perp}),
\end{aligned}
$$

where (1) holds by the definition of data balancing on the joint, (2) holds by the assumption of conditional independence. Therefore, the l.h.s is not a function of $Z$ which establishes conditional independence. $\square$

*Remark* B.4. We would not expect $Y$ and $Z$ to be independent given $X_{\overline{Z}}^{\perp}$ in $Q$ unless $X_{\overline{Z}}^{\perp}$ and $Z$ are independent given $Y$ in $P^t$. From the previous proof, we wrote

$$
Q(Y \mid X_{\overline{Z}}^{\perp}, Z) = \frac{P^t(X_{\overline{Z}}^{\perp} \mid Y, Z) P^t(Y) P^t(Z)}{\sum_Y P^t(X_{\overline{Z}}^{\perp} \mid Y, Z) P^t(Y) P^t(Z)},
$$

so generally we would expect $Q(Y \mid X_{\overline{Z}}^{\perp}, Z)$ to depend on $Z$, which implies $Y \not\!\perp\!\!\!\perp_Q Z \mid X_{\overline{Z}}^{\perp}$, whenever $P^t(X_{\overline{Z}}^{\perp} \mid Y, Z) P^t(Y) P^t(Z)$ depends on $Z$ (i.e. when $X_{\overline{Z}}^{\perp} \not\!\perp\!\!\!\perp_{P^t} Z \mid Y$).

**Proposition B.5** (Equalized odds). $X_{\overline{Z}}^{\perp} \perp\!\!\!\perp_Q Z \mid Y$ *if* $X_{\overline{Z}}^{\perp} \perp\!\!\!\perp_{P^t} Z \mid Y$; *that is data balancing does not disturb independence between* $X_{\overline{Z}}^{\perp}$ *and* $Z$ *given* $Y$ *if* $X_{\overline{Z}}^{\perp}$ *and* $Z$ *are independent given* $Y$ *in the original data distribution* $P^t$.

*Proof.* Let $X_{\overline{Z}}^{\perp} \perp\!\!\!\perp_{P^t} Z \mid Y$. Note that in this case we just need to show that data balancing does not disturb the conditional independence $X_{\overline{Z}}^{\perp} \perp\!\!\!\perp_{P^t} Z \mid Y$ present in the original data (we already had equalized odds in original data). The following derivation demonstrates the claim,

$$
\begin{aligned}
Q(X_{\overline{Z}}^{\perp} \mid Z, Y) &= \frac{Q(X_{\overline{Z}}^{\perp}, Y, Z)}{\sum_{X_{\overline{Z}}^{\perp}} Q(X_{\overline{Z}}^{\perp}, Y, Z)} \\
&\overset{(1)}{=} \frac{P^t(X_{\overline{Z}}^{\perp} \mid Y, Z) P^t(Y) P^t(Z)}{\sum_{X_{\overline{Z}}^{\perp}} P^t(X_{\overline{Z}}^{\perp} \mid Y, Z) P^t(Y) P^t(Z)} \\
&\overset{(2)}{=} \frac{P^t(X_{\overline{Z}}^{\perp} \mid Y) P^t(Y) P^t(Z)}{\sum_{X_{\overline{Z}}^{\perp}} P^t(X_{\overline{Z}}^{\perp} \mid Y) P^t(Y) P^t(Z)} \\
&= P^t(X_{\overline{Z}}^{\perp} \mid Y),
\end{aligned}
$$

where (1) holds by the definition of data balancing and (2) holds by the assumption of conditional independence. Therefore, the l.h.s is not a function of $z$ which establishes conditional independence. $\square$

*Remark* B.6. Similar to the demographic and predictive parity cases, we would expect that, in most cases when $X_{\overline{Z}}^{\perp} \perp\!\!\!\perp_Q Z \mid Y$ holds, it is because $X_{\overline{Z}}^{\perp} \perp\!\!\!\perp_{P^t} Z \mid Y$.

## C  Impact of data balancing on the CBN

Recall that we assumed that $Z$ is discrete, but all the results are easily extended for continuous $Z$.

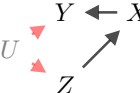

**Proposition 5.1.** *Let* $\langle \mathcal{G}, P^t \rangle$ *be the CBN underlying the data, where* $\mathcal{G}$ *contains an undesired path between* $Z$ *and* $Y$, *and let* $\mathcal{G}^0$ *be a modification of* $\mathcal{G}$ *in which the undesired path has been removed. The distribution* $Q$ *obtained*

*by joint balancing the data to make $Z$ and $Y$ statistically independent, i.e.*
*$Q(Y, X, Z) = P^t(X \mid Y, Z) P^t(Z) P^t(Y)$, might not factorize according to*
$\mathcal{G}^0$.

*Proof.* **Example 1: Causal task with causal and non-causal paths.** Consider $\mathcal{G} = \{Z \rightarrow X \rightarrow Y, Z \leftarrow U \rightarrow Y\}$, for unobserved $U$. We have

$$Q(Y \mid X, Z) = \frac{Q(X, Y, Z)}{\sum_Y Q(X, Y, Z)} = \frac{P^t(X \mid Y, Z) P^t(Z) P^t(Y)}{\sum_Y P^t(X \mid Y, Z) P^t(Z) P^t(Y)} = \frac{P^t(X \mid Z, Y) P^t(Y)}{\sum_Y P^t(X \mid Z, Y) P^t(Y)},$$

where the r.h.s is a function of $Z$ in general as $X$ is not independent of $Y$ given $Z$ in $P^t$. If $Q$ were $\mathcal{G}^0 = \{Z \rightarrow X \rightarrow Y\}$, then $Y \per\!\!\!\perp Z \mid X$ in $Q$. To show the claim it suffices therefore to construct a distribution $P^t$ such that $X$ is not independent of $Y$ given $Z$.

**Example 2: Causal task with non-causal path.** Consider $\mathcal{G} = \{X \rightarrow Y, Z \leftarrow U \rightarrow Y\}$. We have that,

$$Q(X \mid Z) = \frac{\sum_Y Q(X, Y, Z)}{\sum_{Y,X} Q(X, Y, Z)} = \frac{\sum_Y P^t(X \mid Y, Z) P^t(Z) P^t(Y))}{\sum_{Y,X} P^t(X \mid Y, Z) P^t(Z) P^t(Y)} = \sum_Y P^t(X \mid Y, Z) P^t(Y).$$

The r.h.s is a function of $Z$ in general as $X$ is not independent of $Z$ given $Y$ in a distribution $P^t$ consistent with $\mathcal{G}$. Therefore, one may not interpret the mutilated graph $\mathcal{G}^0 = \{X \rightarrow Y\}$ as a correct representation of the conditional independences implied by the balanced distribution $Q$.

**Example 3: Causal task with causal path.** Consider $\mathcal{G} = \{Z \rightarrow X \rightarrow Y\}$. We have that,

$$Q(Y \mid X, Z) = \frac{Q(X, Y, Z)}{\sum_Y Q(X, Y, Z)} = \frac{P^t(X \mid Y, Z) P^t(Z) P^t(Y)}{\sum_Y P^t(X \mid Y, Z) P^t(Z) P^t(Y)} = \frac{P^t(X \mid Z, Y) P^t(Y)}{\sum_Y P^t(X \mid Z, Y) P^t(Y)},$$

The r.h.s is a function of $Z$ in general as $X$ is not independent of $Z$ given $Y$ in $P^t$. Therefore, one may not interpret the mutilated graph $\mathcal{G}^0 = \{Z, X \rightarrow Y\}$ as a correct representation of the conditional independences implied by the balanced distribution $Q$.

**Example 4: Anti-causal task.** Consider $\mathcal{G} = \{Y \rightarrow X, Z \leftarrow U \rightarrow Y, Z \rightarrow W \rightarrow X\}$. We have that,

$$Q(X \mid Z) = \frac{\sum_{Y,W} Q(X, Y, Z, W)}{\sum_{Y,X,W} Q(X, Y, Z, W)} = \frac{\sum_{Y,W} P^t(X, W \mid Y, Z) P^t(Z) P^t(Y))}{\sum_{Y,X,W} P^t(X, W \mid Y, Z) P^t(Z) P^t(Y)} = \sum_Y P^t(X \mid Y, Z) P^t(Y).$$

The r.h.s is a function of $Z$ in general as $X$ is not independent of $Z$ given $Y$ in a distribution $P^t$ consistent with $\mathcal{G}$. Therefore, one may not interpret the mutilated graph $\mathcal{G}' = \{Y \rightarrow X, Z \rightarrow W \rightarrow X\}$ as a correct representation of the conditional independences implied by the balanced distribution $Q$.

$\square$

## C.1   Regularization and data balancing don't always go hand in hand

### C.1.1   Risk-invariance

We first consider the graph in Figure 1(d) and show that $X_{\bar{Z}}^{\perp} \perp\!\!\!\perp Z \mid Y$ in both $Q$, which justifies its use in addition to data balancing, although there might not be a benefit of using both techniques simultaneously (in theory).

**Proposition C.1.** *Consider the graph $\mathcal{G}$ in Figure 1(d). Then $X_{\bar{Z}}^{\perp} \perp\!\!\!\perp Z \mid Y$ in both the training data distribution $P^t$ (consistent with $\mathcal{G}$) and the distribution after balancing, namely $Q$.*

*Proof.* $X_{\bar{Z}}^{\perp} \perp\!\!\!\perp Z \mid Y$ holds in the training data distribution $P^t$ by $d$-separation. For the conditional independence in $Q$, consider the following derivation,

$$Q(X_{\bar{Z}}^{\perp} \mid Y, Z) = \frac{\sum_{X_{Y \wedge Z}} P^t(X_{\bar{Z}}^{\perp}, X_{Y \wedge Z} \mid Z, Y) P^t(Z) P^t(Y)}{\sum_{X_{Y \wedge Z}, X_{\bar{Z}}^{\perp}} P^t(X_{\bar{Z}}^{\perp}, X_{Y \wedge Z} \mid Z, Y) P^t(Z) P^t(Y)}$$
$$= P^t(X_{\bar{Z}}^{\perp} \mid Z, Y) = P^t(X_{\bar{Z}}^{\perp} \mid Y) = g(X_{\bar{Z}}^{\perp} \mid Y)$$

```
import numpy as np
import scipy

# Number of samples.
n = 10000

# Generate binary data with simple data generating model Z -> Y <- X
x = 1*(np.random.normal(size=n) > 0)
u = 1*(np.random.normal(size=n) > 0.3)
y = 1*(x - u + 0.5*np.random.normal(size=n) > 0.5)
z = 1*(u - 0.5*np.random.normal(size=n) > 0.1)

# Marginal of z.
p_z = np.array([np.mean(z==i) for i in z])
# Marginal of y.
p_y = np.array([np.mean(y==i) for i in y])
# Joint of z and y.
p_zy = np.array([np.mean((z==i)&(y==j)) for i, j in zip(z,y)])

# Resampling probabilities
indep_probs = p_z * p_y / p_zy
indep_probs /= np.sum(indep_probs)

# Re-sample according to computed probabilities
indeces = np.random.choice(n, size=n, replace=True, p=indep_probs)
z_bal, x_bal, y_bal = z[indeces], x[indeces], y[indeces]

# Check that Y and Z are independent
# Create contingency table.
contingency_table_bal_zy = scipy.stats.contingency.crosstab(z_bal,y_bal)
# Implement chi squared test.
statistic, pvalue, _, _ = scipy.stats.chi2_contingency(contingency_table_bal_zy)

# Check whether X and Z are independent
contingency_table_bal_xz = scipy.stats.contingency.crosstab(z_bal,x_bal)
statistic, pvalue, _, _ = scipy.stats.chi2_contingency(contingency_table_bal_xz)
```

Figure 7: Python code to assess the impact of balancing in a numerical simulation of graph Figure 1(b).

The r.h.s is not a function of $Z$ and therefore $X_{\overline{Z}}^{\perp} \perp\!\!\!\perp Z \mid Y$ holds in $Q$. $\qquad\square$

However, when considering the graph in Figure 1(b), we introduce a dependence between $X_{\overline{Z}}^{\perp}$ and $Z$, which can be easily checked by the simulation Figure 7 in which we consider the simplified graph $Z \to Y \leftarrow X$. While we are able to obtain the marginal dependence between $Y$ and $Z$ ($\chi^2 : p = 0.34$), we introduce a dependence between $X$ and $Z$ ($\chi^2 : p < 0.0001$).

### C.1.2 When does data-balancing together with regularization lead to fair models?

This section gives several results to analyze the combination of data balancing implemented to generate independence between outcomes $Y$ and sensitive attributes $Z$ and regularization in two variants. First, regularizing to learn representations $W = \phi(X_{\overline{Z}}^{\perp})$ such that $W \perp\!\!\!\perp_Q Z \mid Y$; and second regularizing to learn representations $W = \phi(X_{\overline{Z}}^{\perp})$ such that $W \perp\!\!\!\perp_Q Z$. We write $X_{\overline{Z}}^{\perp} \perp\!\!\!\perp_{P^t} Y$ to state that $X_{\overline{Z}}^{\perp}$ and $Y$ are independent in distribution $P^t$.

**Regularization such that $\phi(X_{\overline{Z}}^{\perp}) \perp\!\!\!\perp Z \mid Y$.**

**Proposition C.2** (Demographic parity). *Balancing and regularization such that $W = \phi(X_{\overline{Z}}^{\perp})$ and $W \perp\!\!\!\perp_Q Z \mid Y$ is sufficient for demographic parity, i.e. $W \perp\!\!\!\perp_Q Z$.*

*Proof.*

$$Q(W \mid Z) = \sum_Y Q(W \mid Z, Y) Q(Y \mid Z) \overset{(1)}{=} \sum_Y Q(W \mid Y) Q(Y) = Q(W),$$

where (1) holds by the assumption of balancing in which $Z \perp\!\!\!\perp_Q Y$ and regularization $W \perp\!\!\!\perp_Q Z \mid Y$. $\qquad\square$

**Proposition C.3** (Predictive parity). *Balancing and regularization such that $W = \phi(X_{\bar{Z}}^{\perp})$ and $W \perp\!\!\!\perp_Q Z \mid Y$ is sufficient for predictive parity, i.e. $Y \perp\!\!\!\perp_Q Z \mid W$.*

*Proof.*

$$Q(Z \mid Y, W) = Q(Z \mid Y) = Q(Z),$$

where both equalities hold by the assumption of balancing in which $Z \perp\!\!\!\perp_Q Y$ and regularization $W \perp\!\!\!\perp_Q Z \mid Y$. $\qquad\square$

**Proposition C.4** (Equalized odds). *Balancing and regularization such that $W = \phi(X_{\bar{Z}}^{\perp})$ and $W \perp\!\!\!\perp_Q Z \mid Y$ is sufficient for equalized odds, i.e. $W \perp\!\!\!\perp_Q Z \mid Y$.*

*Proof.* Regularization induces $W \perp\!\!\!\perp_Q Z \mid Y$ and so equalized odds is satisfied by design. $\qquad\square$

**Remark:** Note that balancing and regularization together are not always necessary, for example the section above shows that balancing on its own can be successful in some cases.

**Regularization such that $\phi(X_{\bar{Z}}^{\perp}) \perp\!\!\!\perp Z$.**

**Proposition C.5** (Demographic parity). *Balancing and regularization such that $W = \phi(X_{\bar{Z}}^{\perp})$ and $W \perp\!\!\!\perp_Q Z$ is sufficient for demographic parity, i.e. $W \perp\!\!\!\perp_Q Z$.*

*Proof.* Regularization induces $W \perp\!\!\!\perp_Q Z$ and so demographic parity is satisfied by design. $\qquad\square$

**Proposition C.6** (Predictive parity). *Balancing and regularization such that $W = \phi(X_{\bar{Z}}^{\perp})$ and $W \perp\!\!\!\perp_Q Z$ is not sufficient for predictive parity, i.e. $Y \perp\!\!\!\perp_Q Z \mid W$ does not hold.*

*Proof.* We give a counter-example. Let $A, B, C$ be three independent variables with values in $\{0, 1\}$. Let $X_{\bar{Z}}^{\perp} = \mathbf{1}\{A = B\}, Y = \mathbf{1}\{A = C\}, Z = \mathbf{1}\{B = C\}$. Let $Q$ be a probability distribution over $(X_{\bar{Z}}^{\perp}, Y, Z)$. In particular, we could imagine $Q$ to be generated after balancing and regularization since $W \perp\!\!\!\perp_Q Z$ and $Y \perp\!\!\!\perp_Q Z$. However, conditioned on $X_{\bar{Z}}^{\perp}$, $Y$ and $Z$ determine each other and so predictive parity does not hold in $Q$. $\qquad\square$

**Proposition C.7** (Equalized odds). *Balancing and regularization such that $W = \phi(X_{\bar{Z}}^{\perp})$ and $W \perp\!\!\!\perp_Q Z$ is not sufficient for equalized odds, i.e. $W \perp\!\!\!\perp_Q Z \mid Y$ does not hold.*

*Proof.* The counter-example above applies. $\qquad\square$

## D Experiments

### D.1 Datasets

This work uses the MNIST [45, 17, http://yann.lecun.com/exdb/mnist/], Amazon reviews [53], ImageNet [16, https://image-net.org/] and CelebA [46, http://mmlab.ie.cuhk.edu.hk/projects/CelebA.html] datasets, which are all openly accessible and can be used for research purposes.

**MNIST semi-synthetic data**: For simplicity, we binarize the digit recognition task to a label $Y \in \{0, 1\}$ according to whether the number in the image is $< 5$ or $\geq 5$ such that $Y$ matches the ground truth with probability $0.98$. The top of the image is replaced by noise coloured in red for $Z = 0$ and blue for $Z = 1$ (see Figure 1). We can relate the confounder and the label such that $95\%$ (resp. $5\%$) of images with $Y = 0$ have a red (resp. blue) noise pattern, while $10\%$ (resp. $90\%$) of the images with $Y = 1$ have a red (resp. blue) pattern, corresponding to our original distribution $P^t$. In this distribution, the marginal distributions of $Y$ and $Z$ are (close to) uniform. We sample $n = 30,000$ samples from $P^t$, as well as a dataset jointly balanced on $Y$ and $Z$ ($Q, n = 30,000$). We also sample test data based on a ground truth $P^0$ generated with $P^0(Z = 0|Y) = 0.5$ ($n = 2,000$). Finally, we generate an $X_{\bar{Z}}^{\perp}$ dataset that contains white instead of colored noise.

**MNIST semi-synthetic data with added confounder**: We add $V$ and $X_V$ to our data generating process where $X_V$ is a green cross either on the left or right of the image, with a fixed vertical

position. The horizontal position of the cross is given by $V$ and $V$ is correlated with $Y$ ($P^t(V = 0|Y = 0) = 0.2$, $P^t(V = 0|Y = 1) = 0.9$). We generate a confounded dataset (95/10) as previously, which we balance jointly on $Y$ and $Z$. We then train 5 replicates of the same architecture, and test our model on $Q$, as well as on the ground truth $P^0$ where $P^0(V = 0|Y = 0) = P^0(V = 0|Y = 1) = P^0(Z = 0|Y = 0) = P^0(Z = 0|Y = 1) = 0.5$.

**MNIST semi-synthetic data, entangled**: We define the color of the noise based on an $OR(Y, Z)$. We define $Q$ by generating samples with $Q(Z = 0|Y = 0) = Q(Z = 0|Y = 1) = 0.5$, while $P^0$ is represented by the disentangled test dataset described above.

**Amazon reviews with confounder**: We refer to Veitch et al. [74] and define a causal task based on Amazon reviews for the clothing category which predicts whether the review was found to be helpful (i.e. obtained 'thumbs up' votes) or not based on the review's text. We generate a random variable $U$ as the unobserved confounder, and define $Y$ as the binary helpfulness label, randomly flipping the label based on $U$ (association: p=0.4). This leads to reviews with $Y = 0$ being more associated with $U = 0$. We define $Z$ as $Z = \lambda * U + (1 - \lambda) * U_2$, where $U_2$ is another random variable distributed uniformly and $\lambda$ is a parameter that controls the relationship between $U$ and $Z$, and by transitivity, between $Z$ and $Y$. In $P^t$, $\lambda$ is selected to be 0.8, leading to a correlation of 0.35 between $Y$ and $Z$. To create $X_Y^\perp$, we add perturbations to the text based on the value of $Z$ that wouldn't (in theory) affect $Y$. We select the words {and, the, you, my, they} and add a suffix 'xxxx' (resp. 'yyyy') when $Z = 0$ (resp. $Z = 1$). Finally, $Y$ is imbalanced, with only 5% of the dataset with $Y = 1$. We hence re-balance the classes before the modelling. This operation is also performed by the joint balancing.

## D.2 Metric definitions and operationalization

Our work focuses on statistical group fairness criteria [5]. These can be translated as independence criteria on the model's predictions.

**Definition D.1** (Demographic parity). A predictor $f(X)$ is said to satisfy demographic parity w.r.t. sensitive attribute $Z$ and distribution $P^t$ if $f(X) \perp\!\!\!\perp_{P^t} Z$.

**Definition D.2** (Predictive parity). A predictor $f(X)$ trained to predict an outcome $Y$ is said to satisfy predictive parity w.r.t. sensitive attribute $Z$ and distribution $P^t$ if $Y \perp\!\!\!\perp_{P^t} Z \,|\, f(X)$.

**Definition D.3** (Equalized odds). A predictor $f(X)$ trained to predict an outcome $Y$ is said to satisfy equalized odds w.r.t. a sensitive attribute $Z$ and distribution $P^t$ if $f(X) \perp\!\!\!\perp_{P^t} Z \,|\, Y$.

In our experiments, we estimate equalized odds as in Alabdulmohsin & Lučić [1]. For this metric, the lower, the better.

$$EO = 0.5 * \max_{z \in \mathcal{Z}} \mathbb{E}_X[f(X)\,|\,Z = z, Y = 0] - \min_{z \in \mathcal{Z}} \mathbb{E}_X[f(X)\,|\,Z = z, Y = 0]$$
$$+ 0.5 * \max_{z \in \mathcal{Z}} \mathbb{E}_X[f(X)\,|\,Z = z, Y = 1] - \min_{z \in \mathcal{Z}} \mathbb{E}_X[f(X)\,|\,Z = z, Y = 1].$$

In terms of robustness metrics, we evaluate a simplified version of risk-invariance by computing model performance on a test set sampled from $P^t$, and contrasting this result with the model's performance on a test set sampled from $P^0$ (when known), or from $Q$. We also estimate worst-group performance [64] as:

$$WG = \min_{z' \in \mathcal{Z}} \mathbb{E}_{X,y}[\mathbb{1}[f(X) = y]\,|\,z = z']$$

An invariant model that is optimal would hence display high performance on both $P^t$ and $P^0/Q$, as well as high worst-group accuracy.

Metrics like risk-invariance or equalized odds provide insights on the model's outputs, but do not probe the model's representation. As we are interested in large-scale models that might be further fine-tuned, it is important to understand whether the model's representation is invariant on $\mathcal{P}$. Defining a representation as $\phi(X)$, we can write $f(X) = h(\phi(X))$ in which we assume the representation to be fixed (i.e. frozen model weights) and $h$ is a learnable function. In Zemel et al. [81], the authors define a fair representation w.r.t. a binary $Z$ as demographic parity on the representation:

$$\mathbb{E}_{X \in X^{Z=z}} \phi(X) = \mathbb{E}_{X \in X^{Z=z'}} \phi(X), \forall z, z' \in \mathcal{Z},$$

where $X^{Z=z}$ corresponds to the samples with $Z = z$. This is equivalent to assessing the 'encoding' of $Z$ in $\phi(X)$, by training a linear layer $h : \phi(X) \to Z$ [27, 8]. Chance level performance of $h(\phi(X))$ would then suggest that the representation is independent of $Z$. In the present work, we estimate the encoding of $Z$ using $P^0$ or $Q$ such that assessing the encoding of $Z$ is equivalent to assessing the encoding of $Z|Y$. Models that encode less of the auxiliary factor $Z$ have been shown to reach a more 'global' optimum compared to models that encode the signal more strongly [independently of whether invariant predictions are obtained 80].

### D.3 Model architectures

We consider multiple architectures in this work, with an attempt to cover different model sizes and characteristics.

- Small convolutional network, similar in spirit to AlexNet [43]. It includes 5 convolution blocks with kernel sizes (4, 3, 2, 2, 2, 2) and output channels (3, 6, 9, 12, 12, 9), with max pooling after each convolution, as well as two dense layers with Relu non-linearity before the output head.
- VGG network [68] with square kernels of size 3, output channels of dimensions (64, 64, 128, 128, 128, 256, 256, 256, 512, 512, 512) and strides (1, 1, 2, 1, 1, 2, 1, 1, 2, 1, 1).
- Vision Transformers [18] of different sizes: ViT-micro (17M parameters), ViT-Tiny (44M), ViT-S (174M) and ViT-B (690M), with the Tiny sizes and up taken from [73].
- For text data, we use the BERT architecture, as defined in TensorFlow Hub.

We use a stochastic gradient descent optimizer with Nesterov momentum of $0.9$ for all models.

#### D.3.1 Hyper-parameter searches

We include a hyper-parameter search over the learning rate (5 values in log-scale between $9e - 5$ and $0.1$) coupled with a batch size search between sizes of 128, 256 and 512 examples. In terms of regularization, the small convolutional network include dropout in the dense layers (search on 0.1, 0.2, 0.3), while VGG includes batch normalization in the dense layers (as per their original implementations). We impose an L2-regularization of $1e - 4$ during training for all architectures.

We note that hyper-parameters did not seem to make a difference on the MNIST results. For VGG, there was a larger variation, as well as a larger variance across multiple seeds.

When performing MMD conditional regularization, we vary the strength of the regularizer in $[0.0, 0.1, 0.2, 0.5, 1., 2., 3., 4., 5., 6., 7., 8., 9., 10.]$, with 5 replicates for each value. To minimize computational expenses, we fix the learning rate to $0.001$, dropout rate to $0.1$ and batch size to $64$ (for downsampled datasets) or $256$.

### D.4 Assets, code and resources

We use the BERT model bert_en_uncased_L-12_H768_A-12 from TensorFlow Hub. All other models are trained from scratch in our code infrastructure written in Python and JAX [7]. The results are then analyzed with Python and the numpy [30], matplotlib [32, https://matplotlib.org/] and pandas [50, https://pandas.pydata.org/] packages. For the small convolutional networks, training was performed with 4 GPUs (V100) and evaluation used 1 GPU per model instance. BERT used 2 Tensor Processing Units (TPUs) for training and 1 TPU for evaluation. For all other models, we used 4 Tensor Processing Units for training and 1 TPU or GPU (P100) for evaluation. We note that, apart from ViT-B and BERT, all experiments could be run on CPU.

## E    Results

### E.1    Failure modes of data balancing with MNIST

**Other confounder**    We notice that correlation between $V$ and $Z$ in $Q$ is decreased ($\rho = -0.16$) compared to $P^t$ ($\rho = -0.60$) but is not null. In addition, we observe that the model relies on $V$ (accuracy on $Q$: $0.769 \pm 0.008$, on $P^0$: $0.647 \pm 0.023$). As a consequence, models trained on $Q$ display a bias w.r.t. $Z$ (see equalized odds and worst group performance).

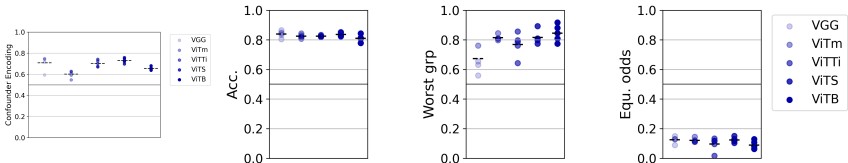

Figure 8: Model encoding, accuracy, worst group accuracy and equalized odds for the VGG architecture, and different sizes of ViT (m: micro, Ti: tiny, S, B) when trained with balanced CelebA data. Each dot is a model replicate, while the dashed line represents the average across replicates.

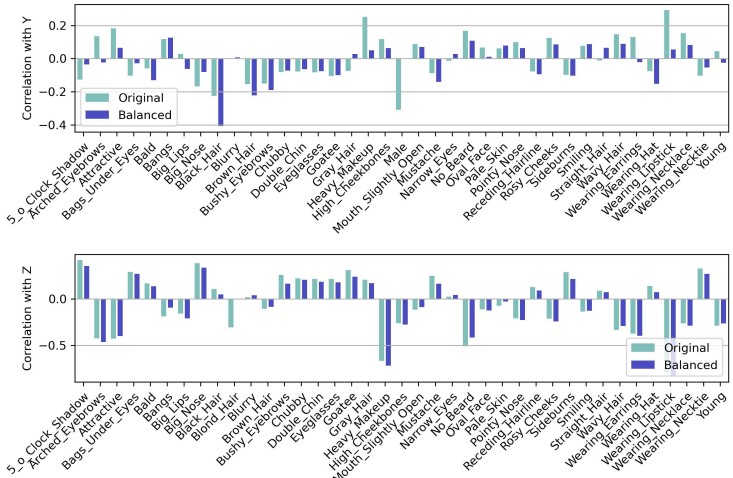

Figure 9: Pearson correlation between each attribute and $Y$ (left), or $Z$ (right) in a sample of the original data (teal), compared to a balanced sample (blue) of the training data.

**Entangled signals** During training, the model reaches $0.903 \pm 0.011$ accuracy on $Q$, but only $0.672 \pm 0.004$ accuracy on $P^0$. Worst-group accuracy is low and equalized odds high, displaying a failure mode of data balancing.

## E.2 Celeb-A

### E.2.1 Model performance

Model encoding and performance across different model sizes is displayed in Figure 8. We show that all models trained on the subsampled data display an encoding of the auxiliary factor $Z$.

### E.2.2 Distinguishing between failure modes

**Correlation patterns in balanced data** We plot the Pearson correlation between $Y$ and all other available attributes (39 in CelebA) in Figure 9 (left), and similarly for $Z$ (right). We note that the correlation that increases most when balancing the data is between $Y$ and the 'black hair' label. As this label has a low correlation with $Z$, this does not seem problematic. We also observe smaller changes in attributes related to hair ('bushy-eyebrows', 'bald') and accessories ('wearing-hat').

## Broader impact

Our work investigates a common mitigation strategy for failures of fairness or robustness in machine learning predictive settings. We aim to clearly highlight when data balancing is promising, and when it fails, hence advancing the field of trustworthy machine learning. As with most papers addressing fairness questions, we acknowledge that our mathematical formulations of fairness criteria might not correspond to the desired societal impact, e.g. in terms of equity. Specific considerations for our work include the use of the CelebA [46] dataset, and in particular the 'is-male' binary label provided.

We acknowledge that a binary characterization of gender is not representative and can be harmful. In addition, it would be desirable to have self-reported instead of perceived gender. Our work considers cases for which auxiliary factors of variation $Z$ are observed at train, test or fine-tuning time. This is a limitation of our investigation, as our insights might not be available when $Z$ is unobserved. This is exemplified by the more difficult case of distinguishing between failure modes without a $P^0$ in the classification of CelebA images.

