# OpenReview forum: "Mind the Graph When Balancing Data for Fairness or Robustness"
_NeurIPS.cc/2024/Conference — NeurIPS 2024 poster_

### Official Review · Reviewer_rHnX · 2024-06-27

**Soundness:** 3
**Presentation:** 3
**Contribution:** 2
**Rating:** 4
**Confidence:** 3

**Summary:**

This paper theoretically studies the applicability of data balancing in achieving fairness or robustness. The paper shows that data balancing may fail depending on the data generation mechanism. The paper introduces conditions for data balancing to produce invariant and optimal models. The paper also shows that data balancing can negatively impact fairness or robustness when combined with regularization strategies.

**Strengths:**

Originality
The paper extends the previous results on data balancing as mitigation for invariant models by considering different failure models and proposes strategies to distinguish between them.

Qualify
The paper offers some valuable insights into data balancing techniques, especially the failure modes of data balancing.

**Weaknesses:**

This theoretical work on data balancing for fairness and robustness in machine learning presents some valuable insights, but falls short in several key areas. The paper's main contribution lies in demonstrating cases where data balancing techniques fail to achieve fairness. However, it does not effectively synthesize these findings into clear principles or rules, lacking a clear, overarching takeaway message. A more effective approach might have been to distill the various cases and propositions into a set of guiding principles or a unified framework for approaching fairness using data balancing.

The authors present several propositions in Section 4, summarizing some of their results. While these propositions offer a formal structure to the paper, the conclusions they provide are relatively intuitive and do not significantly advance the field's understanding of data balancing.

Another limitation of the paper is its focus on negative results. The authors primarily demonstrate scenarios where data balancing techniques are ineffective ("CANNOT results"), without offering constructive solutions to these identified problems. This approach, while valuable for highlighting potential pitfalls, reduces the paper's overall significance and applicability.

**Questions:**

In Assumption 2.3, does “a function of X” mean “a subset of X”?

Can the proposed method be applied to other types of distribution shift, like prior shift or concept shift?

---

> ### Author Rebuttal · Authors · 2024-08-05
>
> We thank the reviewer for their comment and evaluation of our work. We respond to each comment below and provide a new table and figure in the general rebuttal. We hope that these changes provide a more balanced and guiding framing of data balancing for fairness and robustness and that you will consider revising your score if they do.
>
> **W1: lack of guidance** Based on this comment and on the suggestion from Reviewer YT39, we now provide a table including each graph, whether data balancing provides an invariant and optimal model, whether regularization does (and which type), and next steps if they don’t.
>
> We note that section 4 provides conditions that are not dependent on the causal graph of the application and that Proposition 5.1 also holds irrespective of the graph.
>
> **W2: intuitive results** We partially agree with the reviewer that some results might seem intuitive to researchers familiar with the field. We would however argue that most practitioners are not aware of the limitations of data balancing, and that Proposition 5.1 and its consequences on the interaction between data balancing and other mitigation strategies are novel.
>
> To better highlight interesting cases, we slightly modify Figure 1(b) to refer to a purely spurious dependence between $Y$ and $Z$ in the causal case. While it might be intuitive for purely spurious cases to be addressed with joint data balancing, we show that for a causal case this is not a valid assumption. We also add experiments with Amazon reviews to illustrate this result.
>
> Section 3:
>
> **Causal task: helpfulness of reviews with Amazon reviews (Ni et al., 2019)**. Inspired by Veitch et al. (2021), we refer to the causal task of predicting the helpfulness rating of an Amazon review (thumbs up or down, $Y$) from its text ($X$). We add a synthetic factor of variation $Z$ such that words like ‘the' or ‘my' are replaced by ‘thexxxx' and ‘myxxxx' ($Z=0$) or ‘theyyyy' and ‘myyyyy' ($Z=1$). We train a BERT (Devlin et al., 2019) model on a class-balanced version of the data for reference (due to high class imbalance), and compare to a model trained on jointly balanced data, both evaluated on their training distribution and on a distribution $P^0$ with no association.
>
> In this case, jointly balancing improves fairness and risk-invariance, with the model's performance on the training distribution (acc.: $0.574\pm0.016$) being similar to that on $P^0$ (Table 1). This however comes at a high performance cost when compared to the class balanced model's performance on $P$ (acc: $0.658\pm0.015$). Therefore, data balancing might not lead to optimality for this causal task.”
>
> Section 5: We add the following empirical analysis to the discussion on the causal task of Figure 1(b). Please see the pdf added to the main rebuttal for the figure and table.
>
> “We illustrate this result on the Amazon reviews dataset from Section 3 by imposing a marginal MMD regularization $f(X) \perp Z$ during training and evaluating risk-invariance across multiple $P' \in \mathcal{P}$. When training on $P$, we observe that the regularization allows to 'flatten' the curve, such that from medium to high values of MMD regularization, the model is risk-invariant (Figure 4). On the jointly balanced data, medium values of the regularization degrade risk-invariance (see green curves on Figure 4(b)). Overall, model performance is also lower for the models trained on $Q$ compared to models trained on $P$ across test sets from $P' \in \mathcal{P}$, at similar levels of regularization (see Figure 4(c) for MMD=16). This result displays that $X^\perp_Z$ is not a sufficient statistic for $Y$ in $Q$.”
>
> **W3: focus on negative results** Our work does provide success cases, including conditions for these. We also discuss cases in which balancing helps without leading to an invariant model. We hope that with the addition of the guidance table (answer to W1) and the results from Amazon reviews showing that the model is invariant but not optimal, the reviewer will agree that we provide a nuanced and helpful analysis of data balancing for fairness and robustness.
>
> **Q1** A function of $X$ can be a subset of $X$ but can also express a latent variable that does not directly select features, e.g. in an image. We will clarify.
>
> **Q2** In this work, we discuss risk-invariance, optimality and multiple fairness criteria. We select these criteria as they are best suited to capture the effects of undesired dependencies between $Y$ and $Z$ on the model’s outputs. In particular, risk-invariance to correlation shift corresponds to the settings of “spurious correlations” discussed in the field of robustness (e.g. Sagawa et al., 2020). Using a similar framework, one could define conditions for other risk-invariance criteria. This is mentioned in our discussion (line 372). It is however unlikely that joint data balancing on $Y$ and $Z$ would provide invariant models if there is a covariate shift as the balancing does not affect $X$ directly. For concept or prior shifts, the same methodology could be applied, although having a canonical causal graph would help. We will add this comment to the discussion.

---

### Official Review · Reviewer_AEL7 · 2024-07-03

**Soundness:** 3
**Presentation:** 2
**Contribution:** 2
**Rating:** 4
**Confidence:** 3

**Summary:**

The paper analyses training risk invariant models using data balancing. The authors consider the cases in which data balancing might help obtain risk-invariant models and the cases in which data balancing does not achieve the desired effect. The paper also considers the effect of regularization for robust model training, and how that compares to data balancing.

**Strengths:**

The results presented are sound, and the problem considered in the paper is of practical relevance. I liked that the authors tried to analyse the problem on a simple semi-synthetic setup first, before moving onto the real-world data. This provides a nice test-bed to test and motivate the methodology presented.

**Weaknesses:**

1. The paper only considers risk invariance w.r.t. correlation shift. Can this be generalised to other forms of distribution shifts?
2. The paper only considers the question of joint balancing of Y, Z. As the authors show in Proposition 5.1, joint balancing will not effectively remove the undesirable paths in G in all cases. Can the conditions be generalised to other forms of balancing?
3. Obtaining a DAG for a real-world setup involves the decomposition of $X$ (and positing causal relationships between these components). This seems like a non-trivial task in general. The authors don't touch upon the practical aspects of obtaining the DAG in the real world.
4. The paper does not provide a systematic methodology for determining whether data balancing is a good idea for a given real-world dataset. The experiment seems to be using a trial-and-error method to establish the failure modes, but this seems highly impractical.
5. Overall, I think the presentation can be improved. Certain parts of the paper are a bit vague. (See the questions below for more details.)

**Questions:**

1. In line 161, does "performance" correspond to model accuracy? If so, how is the performance on $P^0$ lower than that on P? (where the latter has an overall accuracy of 0.717)
2. How do you try to decode $Z$ from the representation $\phi(X)$? This is an important piece of information that seems to be missing?
3. In Table 1, what is the test distribution on which the accuracy is measured? What is P(Z=1) for example, and is the test distribution of (X, Y, Z) fixed across the different tasks in Table 1? It seems that these test distributions differ across the different tasks. If so, the comparison of accuracy does not seem to be fair.
4. There is no motivation behind the choice of the 4 cases in Figure 1. These seem somewhat arbitrary. Why were these chosen specifically? Is there a systematic way of considering the different cases?
5. Does the model f need to be a neural network necessarily? If so, this should be made clear.
6. What do "penulatimate representation" and "intermediate representation" mean? Are these referring to the hidden layers of the model?
7. I found the paragraph starting on line 270 to be quite vague. For example, the authors talk about "regularization" without explicitly defining the regularization term.

> "If we consider both the purely spurious correlation and the entangled case, we see that regularization and data balancing would have the same effects of blocking any dependence..."

I am not sure I completely understand this statement either. The regularization is a methodology which would potentially modify the distribution of f(X), but the distribution of data (X, Y, Z) would still remain unchanged, so how does regularization change the DAG of data?

In figure 3 and table 2, the accuracy on the original data distribution P should also be logged. In fact, in real-world settings, the practitioners would care about the accuracy on the original data distribution (not Q or P^0).

**Limitations:**

The authors have addressed the limitations of their work in the final section.

---

> ### Author Rebuttal · Authors · 2024-08-05
>
> We thank the reviewer for their comments and questions. We reply to each below, although the format of the rebuttal doesn’t allow us to provide the proposed text amendments. We prioritized answering all comments but would be happy to provide these suggested changes during the discussion. We hope that this alleviates your concerns and would appreciate a revision of our score if it does.
>
> **W1: risk-invariance** In this work, we discuss risk-invariance, optimality and multiple fairness criteria. We select these criteria as they are best suited to capture the effects of undesired dependencies between $Y$ and $Z$ on the model’s outputs. In particular, risk-invariance to correlation shift corresponds to the settings of “spurious correlations” discussed in the field of robustness (e.g. Sagawa et al., 2020). Using a similar framework, one could define conditions for other risk-invariance criteria. This is mentioned in our discussion (line 372). It is however unlikely that joint data balancing on $Y$ and $Z$ would provide invariant models if there is a covariate shift as the balancing does not affect $X$ directly. We will add this comment to the discussion.
>
> **W2: other balancing schemes** Our work focuses on joint balancing as it is commonly used to address undesired dependencies between $Y$ and $Z$. We briefly discuss class and group balancing in the Appendix, but they do not lead to an independence between $Y$ and $Z$ and are not included in further analyses.  It is unlikely that unique conditions can be derived across multiple balancing schemes, especially as these balancing schemes are typically defined based on specific causal graphs (e.g. Kaur et al., 2023, Sun et al., 2023). For instance, Kaur et al. (2023) show that no unique regularization strategy or set of independences can be imposed to obtain an invariant model in an anti-causal case with multiple attributes. We will amend line 383 which discusses this.
>
> **W3: decomposition of $X$** In Veitch et al. (2021), the authors display a causal and an anti-causal graph that decomposes $X$ into its subcomponents. They show that, under certain assumptions, this decomposition always exists. Therefore, when a causal graph of the application exists, the decomposition of $X$ can be performed based on the definitions of the subcomponents of $X$. Similarly, the relationships between these components can be defined by an upper-bound, i.e. only assuming the independences based on the definitions. In our work, we assume specific (in)dependencies between the subcomponents of $X$ to provide an upper-bound on the effectiveness of data balancing. We will add text under Assumption 2.3 to clarify this. We however agree that the decomposition increases complexity when the number of variables increases. In this case, grouping variables can ease the task, as in Kaur et al., 2023.
>
> **W4: application in real-world** In section 6, we posit that we have a partial causal graph, i.e. we make the assumption of an anti-causal task. While we hope to be in the case of Figure 1(a), our results suggest that the graph is more complex. To understand which graph might represent our data generative process, we consider how regularization interplays with potential failures and use these analyses to highlight which graph is most likely. This process does not perform “trial-error” but rather tests specific hypotheses and aims at identifying the most suitable mitigation strategy when the graph is not fully specified. We will make this clearer and reframe Section 6.
>
> **Q1, Q3** All results in Table 1 are assessed on $P^0$. Performance on $P$ is mentioned in the text. We will clarify in the text and table caption. $P^0$ is the same for Figures 1(a) and (d), but includes $V$ for 1(c) as otherwise it would be out-of-distribution due to the absence of $X_V$. We note that we do not compare results with each other, but rather assess whether each case leads to an invariant and optimal model.
>
> **Q2** To assess encoding, we perform transfer learning by fixing the representation $\phi$ and training a new linear layer $h$ that predicts the factor of variation $Z$. All experimental details are in Appendix D, including metrics and their operationalization (D.2). We will make this clear and move materials if possible.
>
> **Q4** These graphs were selected as they represent different cases studied in the literature, albeit sometimes simplified to provide the upper bound for data balancing. This will be added. Beyond the illustration of our results, the specific selection of these graphs does not affect our results as we aim to be graph-independent (e.g. section 4, Proposition 5.1).
>
> **Q5** The model does not require to be a neural network, although the regularization scheme might differ for different architectures.
>
> **Q6** Yes, this will be clarified. We will also add that this condition should be respected by any architecture, although we only formalize it for a specific form of $f(X)$. Please see our response to **W1** from Reviewer YT39 for the full rewrite.
>
> **Q7** We will explicitly relate the recommended “independence between f(X) and Z conditioned on Y” to a regularization term (implemented in experiments with MMD).
>
> **Q8** The regularization does not affect the DAG but would approximate a distribution generated from a DAG in which these paths are blocked. We will clarify in the text.
>
> **Q9** We add the performance on $P$ in tables (see pdf). We note that practitioners who desire fairness and/or robustness criteria evaluate their model on multiple distributions, and that $P^0$ or $Q$ might be more suitable choices as discussed in Dutta et al., 2020.

---

### Official Review · Reviewer_hyP8 · 2024-07-19

**Soundness:** 3
**Presentation:** 3
**Contribution:** 2
**Rating:** 6
**Confidence:** 4

**Summary:**

The paper focuses on the topic of data balancing for fairness and robustness and uses a causal graph as a tool to analyze the effects of data balancing. In the paper, the paper tries to show both the positive impacts and potential pitfalls of data balancing. For the fairness aspect, the paper focuses on the independence between groups and labels in the data. For the robustness aspect, the paper focuses on the lowest risk across a family of target distributions. The paper uses several synthetic and benchmark datasets to support their theoretical analyses on data balancing.

**Strengths:**

* S1. The paper empirically shows both the positive and negative impacts of data balancing, which possibly provides some insights into both the fairness and robustness field.
* S2. The paper also suggests some causal graph-based analyses to understand the effects of data balancing.

**Weaknesses:**

* W1. This paper examines both fairness and robustness, yet the relationship between these two aspects or the reason the paper focuses on these two remains somewhat obscure. The paper could benefit from giving a more in-depth discussion or intuition on why fairness and robustness are susceptible to similar issues. Potential discussions may include 1) how similar analysis can apply to these two different objectives and 2) why the paper focuses on fairness and robustness among various safety criteria.

* W2. While the paper employs synthetic and benchmark datasets to illustrate various failure cases, a more concrete real-world motivating examples would further highlight the importance of addressing these failure cases in both fair and robust training.

* Minor suggestion: It would be better to use full name of CBN in Figure 1.

**Questions:**

The questions are included in the weakness section.

**Limitations:**

The paper has reasonable limitation, future work, and impact section.

---

> ### Author Rebuttal · Authors · 2024-08-05
>
> We thank the reviewer for their comments and suggestions. We respond to each point below:
>
> **W1: fairness and robustness.** Our work focuses on undesired dependencies between $Y$ and $Z$ and on the use of data balancing to mitigate any bias that might result from these dependencies. This focus directly maps to fairness criteria, as well as to robustness to distribution shift in which the additional factor of variation represents the environment. In fact, fairness and robustness criteria can be equivalent in certain cases, as described in [Makar and D’Amour, 2023](http://arxiv.org/abs/2209.09423). Amongst fairness and robustness, we consider multiple, widely used criteria. To the best of our knowledge, other safety criteria do not clearly map to this framing of undesired dependencies (e.g. privacy, accountability, recourse, interpretability, alignment) and do not refer to data balancing. We however note that some terms defined in the Ethics community can be evaluated with fairness and robustness metrics, such as representational harms (equivalent to demographic parity) or stereotypes (e.g. robustness to correlation shifts or equalized odds).
>
> We provide further discussion on the selection of fairness and robustness criteria in the preliminaries:
>
> Section 2.1.: *Due to undesired independencies*, a model may be optimal on $P$ but perform poorly on another distribution of interest $P’(X, Y, Z)$ (e.g. in deployment), and/or might display disparities across subsets of the data (e.g. $P(X, Y |Z = 0)$). *To identify such behavior, various safety criteria have been proposed, with criteria in the fields of *fairness* and *robustness to domain shift* being specifically designed to capture changes in model outputs across distributions*.
>
> **W2: real-world motivating example.** In this work, we focus on simple cases to identify the successes and pitfalls of data balancing. We feel that our synthetic setup illustrates those best, due to their simplicity and the availability of the ground truth causal graph. To further illustrate failure modes in simple cases, we are adding a semi-synthetic version of Amazon reviews, which represents a causal case similar to that of Figure 1(b). Using this dataset, we show that data balancing overall improves fairness and robustness metrics, but does not lead to an optimal model. We further show that performing data balancing hinders the effectiveness of regularization as it leads to models with lower performance across multiple distributions, and introduces variance in the risk for higher values of the regularizing hyper-parameter.
>
> Section 3:
>
> **Causal task: helpfulness of reviews with Amazon reviews (Ni et al., 2019)**. Inspired by Veitch et al. (2021), we refer to the causal task of predicting the helpfulness rating of an Amazon review (thumbs up or down, $Y$) from its text ($X$). We add a synthetic factor of variation $Z$ such that words like 'the' or 'my' are replaced by 'thexxxx' and 'myxxxx' ($Z=0$) or 'theyyyy' and 'myyyyy' ($Z=1$). We train a BERT (Devlin et al., 2019) model on a class-balanced version of the data for reference (due to high class imbalance), and compare to a model trained on jointly balanced data, both evaluated on their training distribution and on a distribution $P^0$ with no association.
>
> In this case, jointly balancing improves fairness and risk-invariance, with the model's performance on the training distribution (acc.: $0.574\pm0.016$) being similar to that on $P^0$ (Table 1). This however comes at a high performance cost when compared to the class balanced model's performance on $P$ (acc: $0.658\pm0.015$). Therefore, data balancing might not lead to optimality for this causal task.
>
> Section 5: We add the following empirical analysis to the discussion on the causal task of Figure 1(b). Please see the pdf added to the main rebuttal for the figure and table.
>
> “We illustrate this result on the Amazon reviews dataset from Section 3 by imposing a marginal MMD regularization $f(X) \perp Z$ during training and evaluating risk-invariance across multiple $P' \in \mathcal{P}$. When training on $P$, we observe that the regularization allows to 'flatten' the curve, such that from medium to high values of MMD regularization, the model is risk-invariant (Figure 4). On the jointly balanced data, medium values of the regularization degrade risk-invariance (see green curves on Figure 4(b)). Overall, model performance is also lower for the models trained on $Q$ compared to models trained on $P$ across test sets from $P' \in \mathcal{P}$, at similar levels of regularization (see Figure 4(c) for MMD=16). This result displays that $X^\perp_Z$ is not a sufficient statistic for $Y$ in $Q$.
>
> **Minor comment** Thank you, we will not use the acronym in the Figure (see table in pdf).

---

### Official Review · Reviewer_YT39 · 2024-07-23

**Soundness:** 3
**Presentation:** 3
**Contribution:** 3
**Rating:** 7
**Confidence:** 4

**Summary:**

The paper studies the role of data balancing e.g. on sensitive attributes or class labels on obtaining fair or robust models. It identifies various causal graphs and corresponding independence conditions under which data balancing is expected to succeed (or may fail) to provide recommendations on when to use (or not to rely on) it. The main contributions are in deriving sufficient conditions for data balancing to yield invariance under correlation shifts and relating the conditions to observations made in prior work through extensive experiments.

**Strengths:**

1. Data balancing is a simple pre-processing method that one may occasionally resort to in practice to achieve fairness or robustness. Explaining when it may not work or when it works is significant.
2. Presentation and visualizations are clear, gives the required background, and explains the results in context of prior work. Jointly treating robustness and fairness in the exposition is a good way to generalize the results across the two problems. I like the organization of the results in successive sections 3-6 which are natural questions one may ask on data balancing.
3. The work discusses the implications when causal graph is not completely specified, features are learned, and tests the hypotheses on real, high-dimensional data (in contrast to causal ML work which typically only tests on tabular data). Experiments are carefully constructed.

**Weaknesses:**

1. The writing can be clarified in places. For instance, I did not find a result which gives both necessary and sufficient condition contrary to the claim in Section 4 - please correct me if wrong.
2. (minor) Results can be summarized more concisely in a figure/table that shows the causal graphs or dependence structures where data balancing is fine and recommendations (e.g. on regularizers) for the other scenarios.
3. (minor) Some more recent works could be discussed. See comments below.

**Questions:**

Please clarify which results are necessary, sufficient, and both.

Minor comments which do not require a response from the authors:

Although not necessary, please consider citing the original works for risk invariance under correlation shift for the claim in line 194, e.g. from Rojas-Carulla 2018 Invariant Models for Causal Transfer Learning https://arxiv.org/abs/1507.05333 or Peters et al. 2016 Causal inference using invariant prediction: identification and confidence intervals https://arxiv.org/abs/1501.01332.
Consider relating the findings to more recent work such as Kaur et al. 2023 Modeling the Data-Generating Process is Necessary for Out-of-Distribution Generalization https://arxiv.org/abs/2206.07837.

Please specify whether it is feasible and how to check if the sufficient conditions hold.

Consider commenting on data augmentation that might achieve the same conditions as data balancing in some scenarios.

I would suggest adding a footnote that the term sufficient statistic differs slightly from its traditional definition e.g. in line 198 it is more appropriate to say that E[Y|X_Z^\perp] is a sufficient statistic for the outcome risk function E[Y|X], not for Y.

---
After the response

Thanks for responding to the my concerns. I do not have any more questions and feel that the work is significant in that it presents a better understanding of data balancing (a commonly-used method in fairness or robustness problems). I have raised my score to 7 as a result.

**Limitations:**

Limitations are acknowledged in good amount of detail.

---

> ### Author Rebuttal · Authors · 2024-08-05
>
> We thank the reviewer for their positive comments on our work. We respond to the comments below:
>
> 1. **Necessary and sufficient conditions**: Thank you for this comment. Our results show that Propositions 4.2 and 4.3 are sufficient, while Proposition 4.4 is necessary. Based on the comment from Reviewer AEL7, we redefine Proposition 4.4 to make it more general, and rework the proof to highlight its necessity. We also rephrase Section 4 to avoid any confusion.
>
> Section 4: “In this section, we introduce *a sufficient condition on the data generative process and a necessary condition on the model representation* that, taken together, lead to a risk-invariant and optimal prediction model after training on $Q$.”
>
> Line 218: “We hence see that when a causal graph of the application is available, Corollary 4.3 can provide indicators on when data balancing might succeed or fail, *with the caveat that Corollary 4.3 is not a necessary condition*.”
>
> Line 220: “While Proposition 4.2 and its corollary provide conditions on the data generating process, prior work has demonstrated that the learning strategy also influences the model's fairness and robustness characteristics. In Proposition 4.2, we assume that the optimal risk-minimizer $f(X):=E_Q [ Y \mid X]=E_Q [ Y \mid X^{\perp}_Z]$ can be found during training a machine learning model $\hat f(X)$. Therefore, $\hat f(X)$ should be of a function class that can represent $E_Q [ Y \mid X^{\perp}_Z]$. Let's consider the special case where $\hat f(X)$ has the form $h(\phi(X))$, in which $h$ is a ``simple'' function of $\phi(X)$. This case would correspond to e.g. the last layers of a neural network or when learning a model based on a representation $\phi(X)$ (e.g. embeddings, transfer learning). We have the following necessary conditions on $h(\phi(X))$:
>
> **Proposition 4.4.**
>  For $\hat f(X) = h(\phi(X))$ to be optimal and risk-invariant w.r.t. $\mathcal{P}$, we require that (i) $h(\phi(X))$ is able to represent $E_{Q}[Y \mid X_Z^\perp]$, and (ii) $h(\phi(X)) = E_Q[Y \mid X_Z^\perp]$ such that $h(\phi(X))$ is only a function of $X_Z^\perp$.
>
> In Proposition 4.4, we require that (i) $h(\phi(X))$ preserves all the information about the expectation of $Y$ that is in $X_Z^\perp$, and that $h(\phi(X))$ only changes with $X_Z^\perp$ and not with $X_Y^\perp$ or $X_{Y \wedge Z}$. Given our assumptions on $h$ and $\phi$, $\phi(X)$ must be disentangled in the sense that the simple function $h$ eliminates any dependence on $X_Y^\perp$ or $X_{Y \wedge Z}$. For example, if $h$ is a linear function, it must be possible to linearly project out all dependence on $X_Y^\perp$ and $X _{Y \wedge Z}$.
> We note that such a representation can be obtained even if the data is entangled, e.g. by dropping modes of variation during training. [...]”
>
>
> Appendix B.1.
>
> Proof:
> Remember that we assume that $\hat f(X)$ takes the form $h(\phi(X))$. If $h(\phi(X))$ cannot represent $E_{Q}[Y \mid X_Z^\perp]$, it is straightforward that $\hat f(X)$ cannot be optimal. Similarly, for $\hat f(X)$ to be risk-invariant w.r.t. $\mathcal{P}$, we need that $h(\phi(X)) = E_Q[Y | X] = E_Q[Y | X_Z^\perp]$ (sufficient statistics). As the right hand side varies only with $X_Z^\perp$, $h(\phi(X))$ can only vary with $X_Z^\perp$ and cannot depend on $X_Y^\perp$ or $X_{Y \wedge Z}$.
>
>
> 2. **Summary of results**: Thank you for this suggestion. Based on this comment and those of other reviewers, we now provide such a table, depicted in the attached pdf.
>
>
> 3. **Recent work**: Thank you for the additional references and other minor suggestions. We have incorporated them in our revision. With regards to Kaur et al., 2023, we note that our Figure 1(c) is a realization of the canonical graph presented in their Figure 2(a). We will add this work to our related works section and include it in possible mitigation strategies when considering multiple attributes. We however note that this work focuses on an anti-causal graph in which $X_c$ (unobserved) is the only cause of $Y$ (hence a sufficient statistics) and studies multiple mitigation strategies. On the other hand, we focus on data balancing and attempt to consider multiple data generative processes.
>
>
> We hope these clarifications alleviate your concerns and that you will consider amending your score to support the publication of this work.

---

> > ### Comment · Reviewer_YT39 · 2024-08-13
> > **Thanks for the response**
> >
> > The response adequately addresses all of my concerns. I particularly appreciate the table summarizing the work and updates to the text on sufficient and necessary conditions. I feel that the work is significant in that it presents a better understanding of data balancing (a commonly-used method in fairness or robustness problems). I have raised my score to 7 as a result.

---

### Author Rebuttal · Authors · 2024-08-05

We thank the reviewers for their constructive comments and suggestions. We have answered each point in detail, but wanted to highlight some of the important changes here for visibility:
- We reframed our findings in a comprehensive table to guide the reader for next steps (see pdf).
- We have added a semi-synthetic dataset based on Amazon reviews to highlight how causal cases can behave unexpectedly (see pdf).
- We have clarified the language throughout the paper and reworked Proposition 4.4 to highlight its necessity, as well as its framing in the context of neural networks.

Our work provides a framework to investigate a popular formulation of data balancing, leading to the detection of failure cases as well as success cases. We discuss both the positive and negative aspects, and aim for generality rather than focusing on a single graph. We complement our theoretical insights with experiments on 3 datasets, in text and vision applications, using multiple architectures.
We hope these changes and our response address most comments and are looking forward to further discussions during the next week.

---

### Decision · Program_Chairs · 2024-09-25

**Decision:**

Accept (poster)

**Comment:**

The paper provides a critical take on data balancing for fairness. The main contribution is in identifying the conditions under which data balancing will/will not work using a causal graph formalism. Reviewers appreciated the technical depth and clarity in presentation. While some of the results may appear obvious in hindsight, the paper does a good job of providing a formal framework and connecting to existing empirical results. Based on the rebuttal responses, I suggest the authors to add the real-world experiment and a discussion of limitations wrt. other kinds of shifts (e.g., covariate shift), and provide guidance on use of data balancing.